

# Variation in Global Chemical Composition of PM$_{2.5}$: Emerging Results from SPARTAN

Graydon Snider[1], Crystal L. Weagle[2], Kalaivani K. Murdymootoo[1], Amanda Ring[1], Yvonne Ritchie[1], Ainsley Walsh[1], Clement Akoshile[3], Nguyen Xuan Anh[4], Jeff Brook[5], Fatimah D. Qonitan[6], Jinlu Dong[7], Derek Griffith[8], Kebin He[7], Brent N. Holben[9], Ralph Kahn[9], Nofel Lagrosas[10], Puji Lestari[6], Zongwei Ma[11], Amit Misra[12], Eduardo J. Quel[13], Abdus Salam[14], Bret Schichtel[15], Lior Segev[16], S.N. Tripathi[12], Chien Wang[17], Chao Yu[18], Qiang Zhang[7], Yuxuan Zhang[7], Michael Brauer[19], Aaron Cohen[20], Mark D. Gibson[21], Yang Liu[18], J. Vanderlei Martins[22], Yinon Rudich[16], Randall V. Martin*[1,2,23]

---

\* Corresponding author email: graydon.snider@dal.ca or randall.martin@dal.ca phone: 902-494-1820, fax: 902-494-5191

*Affiliations*
[1]Department of Physics and Atmospheric Science, Dalhousie University, Halifax, Canada
[2]Department of Chemistry, Dalhousie University, Halifax, Canada
[3]Department of Physics, University of Ilorin, Ilorin, Nigeria
[4]Institute of Geophysics, Vietnam Academy of Science and Technology, Hanoi, Vietnam
[5]Department of Public Health Sciences, University of Toronto, Toronto, Ontario, Canada M5S 1A8
[6] Faculty of Civil and Environmental Engineering, ITB, JL. Ganesha No.10, Bandung 40132, Indonesia
[7]Center for Earth System Science, Tsinghua University, Beijing, China
[8]Council for Scientific and Industrial Research (CSIR), Pretoria, South Africa
[9]Earth Science Division, NASA Goddard Space Flight Center, Greenbelt, Maryland, USA
[10]Manila Observatory, Ateneo de Manila University campus, Quezon City, Philippines
[11] School of Environment, Nanjing University, Nanjing, China.
[12] Center for Environmental Science and Engineering, Indian Institute of Technology Kanpur, India
[13]UNIDEF (CITEDEF-CONICET) Juan B. de la Salle 4397 – B1603ALO Villa Martelli, Buenos Aires, Argentina
[14]Department of Chemistry, University of Dhaka, Dhaka - 1000, Bangladesh
[15] Cooperative Institute for Research in the Atmosphere, Colorado State, Colorado, USA
[16] Department of Earth and Planetary Sciences, Weizmann Institute, Rehovot 76100, Israel
[17] Massachusetts Institute of Technology, Cambridge, MA, 02139, USA
[18]Rollins School of Public Health, Emory University, 1518 Clifton Road NE, Atlanta, GA 30322, United States
[19]School of Population and Public Health, University of British Columbia, Vancouver, British Columbia, Canada
[20]Health Effects Institute, 101 Federal Street Suite 500, Boston, MA 02110-1817, USA
[21]Department of Process Engineering and Applied Science, Dalhousie University, Halifax, Canada,
[22]Department of Physics and Joint Center for Earth Systems Technology, University of Maryland, Baltimore County, Baltimore, Maryland, USA
[23]Harvard-Smithsonian Center for Astrophysics, Cambridge, MA 02138, USA



## Abstract

The Surface PARTiculate mAtter Network (SPARTAN) is a long-term project designed to maximize the chemical and physical information obtained from filter samples collected worldwide. This manuscript discusses the ongoing efforts of SPARTAN to define and quantify major ions and trace metals found in aerosols. Our methods infer the spatial and temporal variability of $PM_{2.5}$ in a cost-effective manner; single filters represent multi-day averaged fine particulate matter ($PM_{2.5}$), while an adjacent nephelometer samples air continuously. SPARTAN instruments are collocated with AERONET to better understand the relationship between ground-level $PM_{2.5}$ and columnar aerosol optical depth (AOD).

We have examined the chemical composition of $PM_{2.5}$ at 12 globally dispersed, densely populated urban locations and a site at Mammoth Cave (US) National Park used as a baseline comparison. Each SPARTAN location has so far been active between the years 2013 and 2015 over 2 to 22 month periods. These sites have collectively gathered over 10 years of quality aerosol data. The major $PM_{2.5}$ constituents across all sites (relative contribution ± SD) were ammonium sulfate (20% ± 10%), crustal material (12% ± 6.2%), black carbon (11% ± 8.4%), ammonium nitrate (4.0% ± 2.8%), sea salt (2.2% ± 1.5%), trace element oxides (0.9% ± 0.6%), water (7.2% ± 3.1%) and residue materials (43% ± 25%).

Analysis of filter samples revealed that several $PM_{2.5}$ chemical components varied by more than an order of magnitude between sites. Ammonium sulfate ranged from 1.1 $\mu$g m$^{-3}$ (Buenos Aires, Argentina) to 17 $\mu$g m$^{-3}$ (Kanpur, India [dry season]). Ammonium nitrate ranged from 0.2 $\mu$g m$^{-3}$ (Mammoth Cave, in summer) to 6.7 $\mu$g m$^{-3}$ (Kanpur, dry season). Equivalent black carbon ranged from 0.7 $\mu$g m$^{-3}$ (Mammoth Cave) to 8 $\mu$g m$^{-3}$ (Dhaka, Bangladesh and Kanpur). Comparison with coincident measurements from the IMPROVE network at Mammoth Cave yielded a high degree of consistency for daily $PM_{2.5}$ ($r^2 = 0.76$, slope = 1.12), daily sulfate ($r^2 = 0.86$, slope = 1.03) and mean fractions of all major $PM_{2.5}$ components (within 6%). Major ions generally agree well with previous studies at the same urban locations (e.g. sulfate fractions agree within 4% for eight out of 11 collocation comparisons). Enhanced anthropogenic dust fractions in large urban areas (e.g. Singapore, Kanpur, Hanoi and Dhaka) were apparent from high Zn:Al ratios.

The expected water contribution to aerosols was calculated via the hygroscopicity parameter $\kappa_v$ for each filter. Mean aggregate values ranged from 0.15 (Manila and Ilorin) to 0.31 (Rehovot); with the latter included a major sulfate event. The all-site parameter mean is 0.19. Chemical composition and water retention in each filter measurement allowed inference of hourly $PM_{2.5}$ at 35% relative humidity by merging with nephelometer measurements. These hourly $PM_{2.5}$ estimates compare favorably with a beta attenuation monitor (MetOne) at the nearby US embassy in Beijing, with a coefficient of variation $r^2 = 0.67$ ($n = 3167$), compared to $r^2 = 0.62$ when $\kappa_v$ was not considered. SPARTAN continues to provide an open-access database of $PM_{2.5}$ compositional filter information and hourly mass collected from a global federation of instruments.



## 1. Introduction

Fine particulate matter with a median aerodynamic diameter less than, or equal to, 2.5 μm (PM$_{2.5}$), is a robust indicator of premature mortality (Chen et al., 2008; Laden et al., 2006). Research on long-term exposure to ambient PM$_{2.5}$ has documented serious adverse health effects, including increased mortality from chronic cardiovascular disease, respiratory disease, and lung cancer (WHO, 2005). Outdoor fine particulate matter (PM$_{2.5}$) is recognized as a significant air pollutant, with an Air Quality Guideline set by the WHO at 10 μg m$^{-3}$ annual average (WHO, 2006). Many regions of the world far exceed these long-term recommendations (Brauer et al., 2015; van Donkelaar et al., 2015), and the impact on health is substantial. The 2013 Global Burden of Disease estimated that outdoor PM$_{2.5}$ caused 2.9 million deaths (3 % of all deaths) and 70 million years of lost healthy life on a global scale (Forouzanfar et al., 2015). Atmospheric aerosol are also the most uncertain agent contributing to radiative forcing of climate change (IPCC, 2013). Aerosol mass and composition also play a critical role in atmospheric visibility (Malm et al. 1994). Additional observations are needed to improve the concentration estimates for PM$_{2.5}$ as global risk factor, and to better understand the chemical components and sources contributing to its formation.

Ground-based observations of PM$_{2.5}$ have insufficient coverage at the global scale to provide assessment of long-term human exposure. Furthermore, no global PM$_{2.5}$ protocol exists for relative humidity (RH) filter equilibration. The U.S. EPA measurements are between 30-40% RH, European measurements are below 50% RH, and different protocols exist elsewhere. Satellite remote sensing offers a promising means of providing an extended temporal record to estimate population exposure to PM$_{2.5}$ on a global scale, and especially for areas with limited ground-level PM$_{2.5}$ measurements (Brauer et al., 2015; van Donkelaar et al., 2015). Even in areas where monitor density is high, satellite-based estimates provide additional useful information on spatial and temporal patterns in air pollution (Kloog et al., 2011, 2013; Lee et al., 2012). However, there are outstanding questions about the accuracy and precision with which ground-level aerosol mass concentrations can be inferred from satellite remote sensing. Standardized PM$_{2.5}$ measurements, collocated with ground-based measurements of aerosol optical depth, are needed to evaluate and improve PM$_{2.5}$ estimates from satellite remote sensing.

Ambient humidity affects the relationship of dry PM$_{2.5}$ with satellite observations of aerosol optical depth. Aerosol water also influences the relationship between dry PM$_{2.5}$ and aerosol scatter. A large body of literature has examined the relationship of aerosol composition with hygroscopicity (e.g. IMPROVE (Hand et al., 2012; IMPROVE, 2015), CSN (Chu, 2004; USEPA, 2015), ISORROPIA (Fountoukis and Nenes, 2007), AIM (Wexler and Clegg, 2002)). More recently Petters and Kreidenweis (2007, 2008, 2013) have developed κ-Kohler theory that assigns individual κ values to all major components, from insoluble crustal materials to sea-salt. Mixed values can then be weighted by local aerosol composition.

The chemical composition of PM$_{2.5}$ also offers valuable information to identify the contributions of specific sources, and to understand aerosol properties and processes that could affect health, climate and atmospheric conditions. Spatial mapping of aerosol type using satellite observations and chemical transport modelling can help elucidate the global exposure burden of fine particulate matter composition (Kahn and Gaitley, 2015; Lelieveld et al., 2015; Patadia et



al., 2013; Philip et al., 2014a), however ground-level sampling remains necessary to evaluate
these estimates and provide quantitative detail. Furthermore, the long-term health impacts of
specific chemical components are not well understood (e.g. Lepeule et al., 2012). There is
insufficient long-term $PM_{2.5}$ characterization information for adequate health impact assessments
of specific aerosol mixtures (e.g. Bell et al., 2007). Urban $PM_{2.5}$ speciation has been conducted in
North America (Hand et al., 2012) and Europe (Putaud et al., 2004, 2010), however there
remains need for a global network that consistently measures $PM_{2.5}$ chemical composition in
densely populated regions.
To meet these sampling needs, the ground-based network SPARTAN (Surface
PARTiculate mAtter Network) is designed to evaluate and enhance satellite-based estimates of
$PM_{2.5}$ by measuring fine particle aerosol concentrations and composition continuously over
multi-year periods at sites where aerosol optical depth is also measured (Holben et al., 1998;
Snider et al., 2015). The network includes air filter sampling and nephelometers that together
provide long-term and hourly $PM_{2.5}$ estimates at low RH (35%).
We discuss the ongoing efforts of the SPARTAN project to quantify major ions and trace
metals found in aerosols worldwide. Section 2 describes the methodology used to infer $PM_{2.5}$
composition. Section 3 describes the implementation of sub-saturated κ-Kohler theory to
estimate aerosol water content based on aerosol compositional information. Section 4 defines
crustal and residue material, black carbon, ammonium nitrate, ammonium sulfate, sea salt, and
trace metal oxides as a function of chemical speciation. Relative aerosol composition is
compared with that reported in available literature to assess the general consistency of our
findings. Section 5 evaluates hourly $PM_{2.5}$ estimates (35% RH) at Beijing with a beta attenuation
monitor at the US Embassy.

## 2. Methodology
SPARTAN has been collecting $PM_{2.5}$ on PTFE filters for at least four months, across 13
SPARTAN sites. Snider et al. (2015) provides an overview of the SPARTAN $PM_{2.5}$ observation
network, the cost-effective sampling methods employed and post sampling instrumental methods
of analysis. Each site utilizes a combination of continuous monitoring by nephelometry and mass
concentration via filter-based sampling. Nephelometer scatter is averaged to hourly intervals at
three wavelengths (457nm, 520nm, 634nm), and converted to 550 nm via a fitted Angstrom
exponent. Total scatter is proportional to $PM_{2.5}$ mass and volume (Chow et al., 2006), hence we
provide dry (35% RH) hourly $PM_{2.5}$ estimates by combining scatter at 550 nm at ambient RH
with filter mass and chemical composition information used to determine water content as
described below.
Briefly, filter-based measurements are collected with an AirPhoton SS4i automated air
sampler. Each sampler houses a removable filter cartridge that protects seven sequentially active
filters plus a field blank. Air samples first pass through a bug screen and then a greased impactor
plate to remove particles larger than $PM_{10}$. Aerosols are collected in sequence on a preweighed
Nuclepore filter membrane (8 μm, SPI) that removes coarse-mode aerosols ($PM_{2.5-10}$), while fine
aerosols ($PM_{2.5}$) are then collected on pre-weighed PTFE filters (2 μm, SKC). For each filter,
sampling is timed at regular, staggered 24-hour intervals throughout a 9-day period. Sampling





ends for each filter at 9 AM when temperatures are low, to reduce loss of semi-volatile
components. Filters are transported inside cartridges to and from measurement sites and between
the central SPARTAN cleanroom and laboratory at Dalhousie University where analysis is
conducted.

Figure 1 shows the locations of operating SPARTAN sites across 11 countries that
operated within the period January 2013 to November 2015. Urban $PM_{2.5}$ concentrations at these
sites span an order of magnitude, from 9 μg m$^{-3}$ (e.g. Atlanta) to nearly 100 μg m$^{-3}$ (Kanpur).
Sites include a variety of geographic regions including partial desert (Ilorin, Rehovot, Kanpur),
coastline (Buenos Aires, Singapore), and developing megacities (Dhaka). Site locations are
designed to sample under a variety of conditions, including biomass burning, (e.g. West Africa
and South America), biofuel emissions (e.g. South Asia), monsoonal conditions (e.g. West
Africa and Southeast Asia), suspended mineral dust (e.g. West Africa and the Middle East) and
urban crustal material. Each SPARTAN site provides a representative example of local and
regional conditions in highly populated areas. The sites of Atlanta and Mammoth Cave are
included for instrument inter-comparison purposes.

### 169  2.1. Filter weighing

Filters (PTFE, capillary) are both pre and post-weighed in triplicate using a Sartorius Ultramicro
balance with 0.1 μg precision. Weighing is performed in a cleanroom facility at 35 ± 5% RH and
20-23°C. A total of 437 quality-controlled filters have been weighed across all SPARTAN sites
(Table 3). The mean weighed collected material on each filter is 100 ± 90 μg. The combined
uncertainty (± 2σ) of repetitive triplicate measurements is 4.0 μg. Filters are subsequently
measured for surface reflectance to obtain black carbon, water-soluble ions, and trace metals.

### 176  2.2. Equivalent Black Carbon (EBC)

We define the equivalent black carbon (EBC) as the black carbon content of PTFE filters derived
via surface reflectance *R* using the Diffusion Systems Smoke Stain Reflectometer EEL 43M
(Quincey et al., 2009) as further discussed in Sect. 4.6.  We use the term equivalent black carbon
following the recommendation of Petzold et al. (2013) for data derived from optical absorption
methods.

### 182  2.3. Trace metals

To maximize the information extracted from the filters, each one is cut in half with a ceramic
blade following approaches similar to Zhang et al. (2013) and Gibson et al. (2009). One filter
half is analyzed for relevant trace metals, i.e. crustal components Zn, Mg, Fe, and Al. We digest
this half by adding it to 3.0 mL of 7% trace metal grade nitric acid, similar to Fang et al. (2015).
The acid/filter combination is boiled at 97°C for 2 hours, and the liquid extract is submitted for
quantitative analysis via inductively coupled plasma mass spectrometry (ICP-MS, Thermo
Scientific X-Series 2).

### 190  2.4. Water soluble ions

Water-soluble ions $NO_3^-$, $SO_4^{2-}$, $NH_4^+$, $Na^+$ are detected using the second filter half. The filter is
spiked with 120 μL of isopropyl alcohol and immersed in 2.9 mL of 18 MΩ Milli-Q water.
Filters and liquid extracts are sonicated together for 25 min before being passed through a 0.45



μm membrane filter to remove larger matrix components. Extractions are analyzed by ion
chromatography (IC) via a Thermo Dionex ICS-1100 instrument (anions) and a Thermo Dionex
ICS-1000 (cations) instrument (Gibson et al., 2013a, 2013b).

## 3. Aerosol hygroscopicity

We apply the single-parameter measure of aerosol hygroscopicity ($\kappa$) developed by Petters
and Kreidenweis (2007, 2008, 2013) to represent the contribution of water uptake by individual
components. The $\kappa$ parameter is defined from 0 (insoluble materials) to greater than 1 for sea
salt. Although initially developed for supersaturated CCN conditions, hygroscopic parameters $\kappa$
have been more recently used in sub-saturated conditions (Chang et al., 2010; Dusek et al., 2011;
Giordano et al., 2013; Hersey et al., 2013). For particle diameters that dominate the mass fraction
of $PM_{2.5}$ (larger than 50 nm), the difference in $\kappa$ between CCN and sub-saturated aerosols is
small (Dusek et al., 2011). The water retention of internal mixtures of aerosol components is
often predicted within experimental error (Kreidenweis et al., 2008). Aged, polarized organic
material, which is a major component of $PM_{2.5}$, shows comparable growth factors both in super-
and sub-saturated regions (Rickards et al., 2013).
The volume hygroscopicity parameter $\kappa_v$ is defined as a function of particle volume $V$ and
water activity $a_w$

$$\frac{1}{a_w} = 1 + \kappa_v \frac{V_d}{V_w} \qquad \text{Eq. 1}$$

where $V_d$ and $V_w$ are the dry particulate matter and water volumes, respectively. To a first-order
approximation $a_w = RH/100$. Aerosol volume growth is related via $\kappa$ and RH by defining $f_v(RH)$
as the humidity-dependent ratio of wet and dry aerosol volume:

$$f_v(RH) \equiv \frac{V_{tot}}{V_d} = \frac{V_d + V_w}{V_d} = a + \kappa_v \frac{RH}{100 - RH} \qquad \text{Eq. 2}$$

Combining the previous equations and relating to a diameter $D$ growth factor ($GF \equiv D/D_d$) yields

$$GF = \left( a + \kappa_v \frac{RH}{100 - RH} \right)^{1/3} \qquad \text{Eq. 3}$$

where $a = 1$, except for sea salt as discussed in Sect. 3.1. Reliable estimates of $\kappa_v$ are available
for individual components (*c.f.* Table 1).
The next sections outline how we apply $\kappa$ to represent mass and volume hygroscopic growth
in major hygroscopic aerosol components. Four species directly contribute to water uptake:
ammonium nitrate ($ANO_3$), ammonium sulfate ($ASO_4$), sea salt (NaCl), and organics. We treat
black carbon (EBC), crustal material (CM), and trace oxides (TEO) as non-hygroscopic. We
evaluated inorganic species' growth curves using the AIM model (Wexler and Clegg, 2002) for
RH = 10 – 90% except for sea salt, which included RH = 0%. Hygroscopic parameters were
matched to modeled fits. Aerosols are treated as internally mixed, without deliquescence or
efflorescence points, as discussed further below.





### 3.1. Inorganic behavior


Figure 2 shows the hygroscopic growth for inorganics. The $\kappa_v$ value of 0.51 for ASO$_4$
best matches the AIM model over RH = 10-90% and is similar to the $GF$-derived $\kappa_v$ = 0.53
estimated by Petters and Kreidenweis (2007). Our AIM-derived ANO$_3$ growth curve is slightly
smaller than ASO$_4$, at $\kappa_v$ = 0.41. Although both ammonium compounds share the same $GF$ = 1.6
at RH = 85% (Sorooshian et al., 2008), ANO$_3$ is less hygroscopic at lower RH.
Sea salt accounts for a small fraction of aerosol mass over land, however its hydrophilic
nature makes it significant for water retention. A 1:1 v/v with water for RH = 0% (Kreidenweis
et al., 2008) yields $a$ = 2 (Eq. 2 and 3). A hygroscopic constant $\kappa_v$ = 1.5 then best fits AIM from
the deliquescence point up to 90% RH.
We follow the widely used convention (e.g. IMPROVE) that PM$_{2.5}$ under variable sub-
saturated RH does not exhibit deliquescent phase transitions. There is compelling evidence to
adopt smooth hygroscopic growth curves. Various experiments show sub-micrometer, internally
mixed aerosols will not deliquesce as readily as pure compounds. For example, Badger et al.
(2006) observed ASO$_4$ aerosol deliquescence is clearly inhibited by the presence of humic acids.
A smooth growth curve has been observed over the range RH = 10 – 85% for ambient aerosols at
Jungfraujoch (Swietlicki et al., 2008). Analysis of submicron aerosol mixtures consisting of
NaCl, ASO$_4$, ANO$_3$, and levoglucosan also showed no apparent phase transition (Svenningsson
et al., 2006).

### 3.2. Organic matter behavior


Identifying a representative organic hygroscopic parameter is challenging, as many volume
growth curves are available based on a variety of laboratory experiments and field campaigns.
Organic composition varies by site, and by season. The Appendix table Al contains a collection
of hygroscopic parameters from the literature. Values for $\kappa_{v,OM}$ range from 0 to 0.2. We choose a
single $\kappa_{v,OM}$ value based on the oxygen/carbon ratio (O:C), which is a function of oxidation,
hence age of the organics. Generally O:C ratios are between 0.2 – 0.8 in urban environments
(Rickards et al., 2013). We select an O:C ratio of 0.5 to represent the populated nature of
SPARTAN sites (e.g. Robinson et al., 2013). This corresponds to an organic parameter of
$\kappa_{v,OM}$ = 0.1 for a variety of organic mixtures (Jimenez et al., 2009).

### 3.3. Aerosol water in multi-component systems


Mass-based hygroscopic water uptake $\kappa_m$ is more convenient than $\kappa_v$ to estimate water
retention in gravimetric analysis. The parameters $\kappa_v$ and $\kappa_m$ are related by water-normalized
density, $\kappa_{m,X} = \kappa_{v,X}/\rho_X$. Table 1 contains $\kappa_v$ values identified for major aerosol chemical
components and densities. For a multi-component system we estimate aerosol water mass using a
mass-weighted combination of $\kappa_m$ values:

$$\kappa_{m,tot} = \frac{1}{M} \sum_X m_X \kappa_{m,X} \qquad \text{Eq. 4}$$

Mass calculations are used to determine residue aerosol mass as described in Sect 4.9.
Estimates of total water uptake by volume are applied to aerosol light scatter in Sect. 5. The





volume parameter $\kappa_{v,tot}$ is similarly determined by a linear combination of volume-weighted
components $X$ (e.g. Bezantakos et al., 2013):

$$\kappa_{v,tot} = \frac{1}{V} \sum_X v_X \kappa_{v,X} \qquad \text{Eq. 5}$$

The hygroscopic growth of ASO$_4$ and organic mixtures are treated as linear combinations of pure
compounds (Robinson et al., 2013). Errors in aerosol water uptake are less significant in mixtures
than for individual species due to dilution effects (Kreidenweis et al., 2008). For ambient aerosols,
$\kappa_{v,tot}$ usually lies between 0.14 and 0.39 (Carrico et al., 2010).

## 4. PM$_{2.5}$ aerosol composition

Section 2 defines the methodology of basic chemical species obtained in SPARTAN filters.
Section 4 defines the chemical assumptions made when compiled into Figure 1. Each component
is discussed in turn below. Table 2 contains a summary of equations and accompanying
references used to quantify SPARTAN PM$_{2.5}$ chemical composition.

### 4.1. Sea Salt

We take 10% of [Al] to be associated with Na and remove this crustal sodium component
(Remoundaki et al., 2013). Sea salt is then represented as 2.54[Na$^+$]$_{ss}$ to account for the
associated [Cl$^-$] (Malm et al., 1994).

### 4.2. Ammonium nitrate (ANO$_3$)

We assume that all nitrate is neutralized by ammonium as NH$_4$NO$_3$. The corresponding mass of
ANO$_3$ is a 1:1 molar ratio of NH$_4$:NO$_3$, or 1.29[NO$_3^-$] based on molecular weight.

### 4.3. Sodium sulfate (Na$_2$SO$_4$)

Sodium sulfate is treated as a fraction of measured sodium, 0.18[Na$^+$]; however, it contributes
negligibly to total aerosol mass (< 0.1%) at all sites.

### 4.4. Ammonium sulfate (ASO$_4$)

Ammonium not associated with nitrate, and sulfate not associated with sodium, are assumed to
mutually associate as a mixture of NH$_4$HSO$_4$ and (NH$_4$)$_2$SO$_4$.

### 4.5. Crustal material (CM)

Crustal material consists of re-suspended road dust, desert dust, soil, and sand. Following the
elemental composition of natural dust by Wang (2015), we generalize that natural CM consists
of 10% [Al + Fe + Mg]. Aluminum, iron, and magnesium are chosen due to their collectively
consistent composition in dust (predominantly natural origin) and frequency above detection
limit (> 95%). Silicon is not available for dust analysis. Titanium was found not to contribute
significantly (< 1%) to CM mass.




### 4.6. Equivalent Black Carbon (EBC)

The amount of EBC carbon (µg) is logarithmically related to concentration, as determined by
relative surface reflectance $R/R_0$. For a given exposed filter area, absorption cross-section and
light path, reflectance is related to concentration via

$$[EBC] = \frac{-A}{qv} \ln\left(\frac{R}{R_0}\right)$$
Eq. 6

where $v$ is volume of air (0.9 to 5.8 $m^3$), $A$ is the filter surface area (3.1 $cm^2$), and $q$ is the product
of the effective reflectivity path $p$ and mass-specific absorption cross section $\sigma_{SSR}$ ($cm^2$/µg). The
absorption coefficient $\sigma_{SSR}$ used here is 0.06 $cm^2$/µg based on prior literature (Barnard et al.,
2008; Bond and Bergstrom, 2006), adjusted to the 620 nm detection peak of the SSR. The
effective light path $p$ here is taken to be 1.5 for our thick PTFE filters (e.g. Taha et al., 2007). We
treat water uptake by EBC as negligible.

### 4.7. Trace elemental oxides (TEO)

Trace elemental oxides are the summation of oxides for all measured ICP trace elements, and
make up a negligible portion of total mass (< 1%). We include these concentrations for
completeness while assuming negligible water uptake by TEO.

### 4.8. Particle-bound water (PBW) associated with inorganics

We estimate the water-mass uptake for the inorganic fraction of aerosols species sea salt (NaCl),
ammonium nitrate (ANO$_3$) and ammonium sulfate (ASO$_4$). The mass of particle-bound water
(PBW) associated with species $X$ is

$$PBW_X = [X]\kappa_{m,X}\frac{RH}{100 - RH}$$
Eq. 7

The total mass of inorganic (IN) PBW is then $PBW_{IN} = \sum_X PBW_X$.

### 4.9. Residue matter (RM)

Residue matter at our weighing conditions of 35 ± 5 % RH ($RM_{35\%}$) is estimated by subtracting
dry inorganic mass, $[IN] = \Sigma[X]$, and its associated water from $PM_{2.5}$:

$$RM_{35\%} = PM_{2.5,35\%} - [IN] - [PBW_{IN}]$$
Eq. 8

Negative $RM_{35\%}$ values are retained if reconstructed inorganic mass at 35% RH exceeds total
$PM_{2.5}$ by less than 10%, otherwise values are flagged and excluded from the mass average.
Negative values occur, on average, 2% of the time. Water-free RM (0% RH) is estimated by
subtracting organic-associated PBW using an estimated hygroscopic parameter $\kappa_{m,RM} = 0.07$
(Table 1).

### 4.10.    Overview of PM$_{2.5}$ mass speciation

Table 3 and Figure 1 contain the resulting $PM_{2.5}$ mass and composition at SPARTAN sites.
The mean SPARTAN composition over all sampling sites in descending concentration is 43%
RM, 20% ASO$_4$, 12% CM, 11% EBC, 4% ANO$_3$, 2% NaCl and 1% TEO.





333   There is significant variation of relative and absolute speciation from long-term averages.
334  Concentrations of $PM_{2.5}$ span an order of magnitude, from 9 µg m$^{-3}$ (Atlanta, winter-spring) to 97
335  µg m$^{-3}$ (Kanpur, dry season). $ASO_4$ concentrations range from 1 µg m$^{-3}$ (Buenos Aires, summer)
336  to 17 µg m$^{-3}$ (Kanpur, dry season). The fraction of sulfate in $PM_{2.5}$ exhibits much weaker spatial
337  variation (10-30%). Increases in $ASO_4$ coincide with increases in total $PM_{2.5}$ but less pronounced
338  fractional increases. Hence locations with enhanced sulfate sources tend to have enhancements in
339  sources of other PM components.
341   $ANO_3$ concentrations exhibit even larger spatial heterogeneity than sulfate. Absolute values
342  range from 0.2 µg m$^{-3}$ (Mammoth Cave, summer) to above 6 µg m$^{-3}$ (Kanpur, dry season).
343  Corresponding mass fractions are 7-8 % in Kanpur, Beijing, and Buenos Aires, and below 2% in
344  Bandung. This heterogeneity reflects large spatial and temporal variation in $NH_3$ and $NO_x$
345  sources. There were noticeable increases in $ANO_3$ during wintertime periods in Beijing, Kanpur,
346  and Dhaka, coinciding with lower temperatures.
348   CM concentrations span an order of magnitude from 0.9 µg m$^{-3}$ (Atlanta) to 13 µg m$^{-3}$
349  (Beijing). The fraction of CM in $PM_{2.5}$ exhibits pronounced variation (4-24%). Except during
350  dust storms, CM does not show clear patterns of temporal or regional variation. This could be
351  explained by the significant, non-seasonal contribution from road dust. Anthropogenic dust may
352  account for over 80% of CM in regions with urban traffic and agriculture (Huang et al., 2015).
354   We used Zn:Al ratios to assess the relative importance of local road dust (*c.f.* Table 3).
355  Aluminum is mostly natural in origin (Zhang et al., 2006) whereas Zn is primarily from tire wear
356  (Begum et al., 2010; Councell et al., 2004). For example, ratios are above 2 for Dhaka and
357  Hanoi, but less than 0.5 for Ilorin, Mammoth Cave, Beijing, Atlanta, and Buenos Aires. In fine-
358  mode aerosols, the ratio tends to be highest in large cities distant from natural CM. In coarse-
359  mode aerosols, a low Zn:Al ratio (< 0.1) indicates that aerosol mass is dominated by regional
360  dust.
362   Of the remaining PM components, EBC is highly heterogeneous. Absolute values range from
363  0.9 µg m$^{-3}$ (Atlanta) to 8 µg m$^{-3}$ (Dhaka and Kanpur). Mass fractions of EBC ranged from 3%
364  (Singapore) to 24% (Manila). Trace element oxide (TEO) material is mainly composed of Zn,
365  Pb, Ni, Cu, and Ba, hence also derived mainly from anthropogenic sources. TEO contributes
366  negligibly to total mass (1%), as expected. Sea salt remains a consistently small contributor (2%)
367  to total mass, except for Buenos Aires and Rehovot (6%) due to coastal winds. Particle-bound
368  water (PBW) mass, at 35% humidity, is determined from the growth parameter $\kappa_m$ ($\kappa_v/\rho$, *c.f.* Eq.
369  7 and Tables 1 & 2). PBW is a function of $ASO_4$, $ANO_3$, and sea salt, with a mass contribution
370  similar to EBC (7%). At low humidity the combined mass of $ANO_3$, EBC, TEO, sea salt, and
371  PBW account for 15-35 % of aerosol mass.
373   The RM fraction is implicitly understood to be the organic aerosol mass fraction. RM is
374  inferred from mass reconstruction of inorganic compounds, PBW, and total filter-weighed mass.
375  In terms of relative composition, RM spans a factor of two, from 30% mass in Buenos Aires to
376  almost 60% in Kanpur, and averages about half the total mass of $PM_{2.5}$. Temporal changes in
377  RM tend to coincide with increases in $ASO_4$, with an all-site $r^2 = 0.92$. Although RM is not fully
378  independent of $ASO_4$, this relationship may imply common sources.





We also interpreted the abundance of water-soluble potassium K relative to Al as an indicator
of wood smoke (Munchak et al., 2011). K:Al ratios averaged over each site range from 1.4
(Mammoth Cave) to 19.9 (Singapore). Singapore was downwind of significant Indonesian forest
fire smoke during our Aug-Sept 2015 sampling period. Combustion activity is also apparent in
Kanpur (K:Al = 14.0), whereas Ilorin, Buenos Aires, and Atlanta show less combustion activity
with ratios below 3.
We investigated general compositional correlations between and within aerosol modes.
Coarse and fine mode mass fractions are approximately equal. The all-site mean of $PM_{2.5}/PM_{10}$
fraction is 0.49, with fractions ranging from below 0.40 (Hanoi, Buenos Aires, and Manila) to
above 0.55 (e.g. Bandung, Kanpur, Atlanta, Mammoth Cave, and Singapore). High fine-mode
fraction are therefore not necessarily an indicator of high absolute $PM_{2.5}$. The two modes were
also temporally correlated, though sometimes weakly, from $r^2 = 0.15$ (Hanoi) to $r^2 = 0.76$
(Rehovot). Within $PM_{2.5}$, we observe strong temporal correlations between sulfate and
ammonium ($r^2 = 0.72 - 0.99$). Nitrate and ammonium are less consistently related, ranging from
high values in Kanpur ($r^2 = 0.72$) and Beijing ($r^2 = 0.58$), to weaker values in Ilorin and Manila
($r^2 = 0.11$). The strength of correlations could be influenced by excess ammonium relative to
sulfate. The $[NH_4^+]/[SO_4^{2-}]$ ratio in $PM_{2.5}$ is 2.6 in Kanpur and 1.3 in Ilorin.
### 4.11.     Collocation overview
We compare SPARTAN $PM_{2.5}$ speciation with previous measurements available from the
literature, focusing on relative $PM_{2.5}$ composition of major species from collocated studies within
the last 10 years. TEO is omitted due to lack of significant mass contribution. Aerosol water
content is also omitted as it was not directly measured in any of these collocation studies. If not
provided, CM is treated as defined in Sect 4.5 where possible. Organic mass (OM) to organic
carbon (OC) ratios are from Philip et al. (2014b) with updates from Canagaratna et al. (2015).
Figure 3 provides an overview of the comparison studies organized by SPARTAN data
availability. Only sampling at Mammoth Cave sampling was temporally coincident with the
comparison data. SPARTAN compositional information is generally consistent with previous
studies, considering inter-annual chemical variation and measurement uncertainty. For example,
both SPARTAN and comparative studies find that $PM_{2.5}$ is composed of between 10-30% $ASO_4$
and 5-20% CM for all sites. SPARTAN EBC mass fraction generally matches within 5
percentage points of collocated studies, except for Bandung and Kanpur. SPARTAN and prior
studies find that $ANO_3$ is usually a small fraction of total mass, except at Beijing and Kanpur
(7%) due to their high agricultural and industrial activity. All studies find that sea-salt is below
3% of total mass. SPARTAN-derived RM has potentially the largest potential error, yet typically
is consistent with the combined organic and unknown masses of other studies. SPARTAN
measurements support the expectation that RM is predominantly organic.
Below we discuss each site in more detail. We also examined how our chemical composition
from a global array of sites relate to local anthropogenic activities and surrounding area.
References to land type at specific sites are derived from Latham et al. (2014), unless otherwise
indicated.






### 4.12. Individual site characteristics

#### *4.12.1 Beijing, China (n = 100)*

Beijing has attracted considerable attention for its air pollution (Chen et al., 2013). Agricultural areas to the west and the Gobi Desert to the north surround the city's 19 million dwellers. The SPARTAN air sampler is located on the Tsinghua University campus, 15 km from the downtown center. This is our longest-running site, with over two years of near-continuous sampling to date. It reports the third-highest $PM_{2.5}$, at 69 µg m$^{-3}$, the third highest $ASO_4$ (12 µg m$^{-3}$) and the highest CM (13 µg m$^{-3}$) of all sites. The high $ANO_3$ (4.7 µg m$^{-3}$) reflects significant urban $NO_x$ near agricultural $NH_3$ sources. $ANO_3$ values were highest during winter, as expected from ammonium-nitrate thermodynamics. A high CM component in the springtime reflects regional, natural CM sources. The mean $PM_{2.5}$ Zn:Al ratio is lower than other large cities (0.45) likely due to significant natural dust sources. The lowest coarse-mode Zn:Al mass ratios are observed in April 2014 (0.07) and April 2015 (0.06) during the annual Yellow dust storm season. This is balanced by urban dust sources, in agreement with Lin et al. (2015) who found high CM in industrial areas of Beijing.

*Beijing Comparison:* Relative masses in Beijing compare well with previous studies. SPARTAN $ASO_4$ (19%) is close to Yang et al. (2011) (17%) and Oanh et al. (2006) (20%) and the RM of 42% was similar to their combined OM (33 and 29%) and unknown fractions (10 and 24%). SPARTAN $ANO_3$ concentrations (7%) are lower than either previous study (11-12 %), possibly due to volatilization losses, but still relatively large. CM is comparable to Yang et al. (2011) (21% vs. 19%), but higher than Oanh et al. (2006) (5%), potentially due to different definitions.

#### *4.12.2 Bandung, Indonesia (n = 71)*

Bandung is located inland on western Java surrounded by a volcanic mountain range and agriculture (e.g. tea plantations). The sampler is located on the ITB campus, 5 km north of the city center. Almost two years of sampling have resulted in a mean $PM_{2.5}$ concentration of 31 µg m$^{-3}$. Sea salt is low at this elevated (826 m) inland site. $ANO_3$ and CM levels are also low, but RM is moderately high compared with other sites, at 51%, which could be explained by large amounts of vegetative burning. Organic $PM_{2.5}$ mass fractions can rise above 70% during combustion episodes (Fujii et al., 2014). Volcanic sources of sulfur, in addition to industrial sources, may explain the relatively higher $ASO_4$ compared with Manila or Dhaka (Lestari and Mauliadi, 2009). Influxes of volcanic dust from the Sinabang volcano from August – September 2014 (2000 km northwest of Bandung) could explain why coarse-mode Zn:Al ratios drop to 0.09 for this period compared to the annual mean coarse-mode ratio of 0.21.

*Bandung Collocation:* Bandung is a volcanically active area, so that composition, in particular $ASO_4$, differs due to naturally variable circumstances. SPARTAN $ASO_4$ (20%) is higher than Lestari and Mauliadi (2009) (4%) while more consistent with Oanh et al. (2006). SPARTAN EBC (14%) is less than either previous study (19% and 25%) and the more recent analysis of 19% BC (Santoso et al., 2013). SPARTAN $ANO_3$ is less than 2% relative mass, lower than measured by Oanh et al. (2006) (13%) but similar to Lestari and Mauliadi (2009).





Both of the earlier studies show lower RM fractions (36%, and 42%) compared with our 54%
RM.
### 4.12.3 Manila, Philippines (n = 56)
Manila is a coastal city located in Manila Bay, adjacent to the South China Sea and
surrounded by mountains. The sampling station, located at the Manila Observatory, is about 40
m higher in altitude than the central city. The $PM_{2.5}$ concentrations at the observatory (17 μg m$^{-3}$)
are expected to be lower than in the main city, but still influenced by vehicular traffic, fuel
combustion and industry (Cohen et al., 2009). Compared to the all-site average, the CM fraction
in Manila is typical (11%), but black carbon is twice as great (25%). The high EBC agrees with
previous observations, attributable to a relatively high use of diesel engines (Cohen et al., 2002).
*Manila Collocation:* SPARTAN fractions of $ASO_4$ and EBC are similar to Cohen et al.
(2009). Our RM (44%) is lower than OM (57%), whereas SPARTAN CM was greater than
Cohen et al. (2009). These differences could reflect sampling differences, or emission
changes over the last decade.
### 4.12.4 Dhaka, Bangladesh (n = 41)
Dhaka is a densely populated city (17,000 persons/km$^2$) in a densely populated country
(1,100 persons/km$^2$). The sampler is situated in the heart of downtown Dhaka, on the University
of Dhaka rooftop, and is influenced by air masses from the Indo Gangetic Plain (Begum et al.,
2012). More than half the country is used for agricultural purposes (Ahmed, 2013). Local
contributing $PM_{2.5}$ sources include coal and biomass burning, and heavy road traffic combustion
products and dust (Begum et al., 2010, 2012). $PM_{2.5}$ concentrations are the fourth highest of any
SPARTAN site, at 52 μg m$^{-3}$. Dhaka has the second-highest absolute EBC of any site, at 8.0 μg
m$^{-3}$. A high EBC can be explained by the abundance of truck diesel engines (Begum et al.,
2012). We estimate 41% of $PM_{2.5}$ in Dhaka is RM. Crop or bush burning on both a local and
regional scales contribute significantly to organics (Begum et al., 2012). The high mean $PM_{2.5}$
Zn:Al ratio of 2.48 reflects a large contribution from urban traffic.
### 4.12.5 Ilorin, Nigeria (n = 40)
Ilorin is located in a rural area with low-level agriculture and shrub vegetation; The sampler
is sited on the university campus, 15 km east of the city of 500,000 people. Aerosol loadings
have seasonal cycles from agricultural burning events and dust storms (Generoso et al., 2003).
The RM accounted for two thirds of total mass, among the largest, influenced by biomass
burning. There is evidence of biomass burning in the $PM_{2.5}$ peak in late spring 2014, and again in
2015. Lower $ASO_4$ (12%) compared to other SPARTAN sites reflects the sparse surrounding
industry. CM levels are comparable to other locations, except during dust storms. During a dust
storm (between April 14$^{th}$ - May 2$^{nd}$ 2015), CM explained 65% of total mass, 3% came from
combined sea salt, $ANO_3$ and $ASO_4$, leaving a significant fraction (50%) assigned to RM. The
$PM_c$ Zn:Al ratio during the storm was 0.01, compared with 0.25 during non-storm days.
### 4.12.6 Kanpur, India (n = 33)
Kanpur is a city of 2.5 million people. The sampler is located at the IIT Kanpur campus
airstrip, about 10 km northwest of the city. Kanpur lies in the Indo-Gangetic Plain, where
massive river floodplains are used for agricultural and industrial activity (Ram et al., 2012). We



sampled from December 2013 – May 2014, and September-November 2014, capturing one dry
season. SPARTAN-measured $PM_{2.5}$ for this period was 97 µg m$^{-3}$, the highest of any SPARTAN
site, of which 55% is RM, 18% $ASO_4$, and 7% $ANO_3$. The absolute values of all three
components are also the highest among those measured. Molar $[NH_4^+]:[SO_4^{2-}]$ ratios are higher
in Kanpur (2.6) than elsewhere as well. High background ammonia has been observed in the
region from satellite (e.g. Clarisse et al., 2009), and would explain the high levels of $ANO_3$.
Wood smoke is apparent from the high K:Al ratio (> 10), associated with organic matter burning
during winter dry months. We detected significant Zn concentrations (Zn:Al = 1.4), in agreement
with Misra et al. (2014), who observed a tripling of zinc during pollution-sourced episodes.
*Kanpur Collocation:* SPARTAN $PM_{2.5}$ concentrations, as well as RM, reach a maximum
during the month of December. This is consistent with recent work (Villalobos et al., 2015),
who attribute this increase to agricultural burning and stagnant air. Relative fractions among
the major species CM, salt, $ASO_4$ & $ANO_3$ all match well with previous studies (Behera and
Sharma, 2010; Chakraborty et al., 2015; Ram et al., 2012) that also sampled during winter
dry seasons. Chakraborty et al. (2015) measured 70% organic mass composition and found a
combined mass of 28% for $ASO_4 + ANO_3$ compared to SPARTAN mass (26%). SPARTAN
$ASO_4$ (19%) compares well to 13% for Ram et al. (2012) and 18% for Behera and Sharma
(2010), and $ANO_3$ (7.4%) is close to previous values (6.1% and 6.6%). SPARTAN EBC is
slightly overestimated, by 4-6%. SPARTAN CM (5%) is lower than Behera and Sharma
(2010) (10%). Notably the combined OM + unknown fractions from these previous two
studies account for two thirds of aerosol mass, similar to our 60% RM estimate.
**4.12.10 Buenos Aires, Argentina (n = 21)**
Buenos Aires has a metropolitan population of 12 million. SPARTAN instruments are
located on the urban CITEDEF campus 20 km west of the central downtown. The megacity, the
southernmost in our study, is surrounded by grassland and farming on the west and the Atlantic
Ocean on the east. The latter explains the relatively high proportion (6%) of sea salt. Total $PM_{2.5}$
(10 µg m$^{-3}$) and relative RM (27%) are low compared with other large metropolitan areas, likely
due to clean maritime air. In addition to sea salt and natural CM, the contribution of EBC (11%),
which could be explained by significant local truck diesel combustion (Jasan et al., 2009).
**4.12.7 Mammoth Cave NP, US (n = 19)**
The Mammoth Cave sampling site straddles National Park mountainous terrain to the north
and east, with farmland to the south and west. It is about 35 km from the closest town, Bowling
Green, KY, with about 50,000 residents. Sources of PM are expected to be non-local, hence we
consider it our 'background' site.
*Mammoth Cave National Park Collocation*: This temporary SPARTAN site was deployed
for comparison with the IMPROVE network station (IMPROVE, 2015). Unique among our
sites, sampling was temporally coincident with IMPROVE's 1-in-3 day regimen. We
obtained quality-controlled samples from June-August 2014. Temporal variation in daily
values is consistent with IMPROVE for sulfate (r$^2$ = 0.86, slope = 1.03) and total mass (r$^2$ =
0.76, slope = 1.12). Differences between IMPROVE and SPARTAN are small for $ASO_4$
(36% vs. 30%), $ANO_3$ (2.4% vs. 1.1%), CM (11% vs. 7%), and EBC (4% vs. 3%). The





combined OM + unknown + water fraction IMPROVE was 51%, similar to the SPARTAN
RM mass fraction of 47%.
***4.12.8 Rehovot, Israel (n = 19)***
Rehovot is located on a four-story rooftop on the Weizmann Institute campus, 11 km from
the Mediterranean Sea and 20 km south of Tel Aviv. The city is surrounded by semi-arid, mixed-
use cropland, and the region experiences occasional Saharan desert dust outbreaks. Typical $PM_{2.5}$
concentrations are low (15 μg m$^{-3}$), with the composition in Rehovot consisting of 32% $ASO_4$,
18% RM and 13% CM. The RM fraction is smaller in Rehovot than at other SPARTAN sites
(22% total mass). Aerosol sources in Israel include agriculture, desert dust, traffic and coal-based
power plants (Graham et al., 2004). Relative sodium concentrations are significant in Rehovot
(6%), similar to Buenos Aires and Ilorin, and may include a contribution from dust. A spike in
$ASO_4$ concentrations occurred during the Lag Ba'Omer festival (May 7-18, 2015), during which
a large number of bonfires were lit nearby. During the festival, over 75% of total aerosol mass
came from $ASO_4$ + $ANO_3$, leading to a brief doubling of the hygroscopic parameter $\kappa_v$. We
observed a K:Al ratio of 38 for May 6$^{th}$ of the festival, the highest for any single filter.
*Saharan dust storm:* We had the opportunity to measure a severe dust storm in Rehovot from a
filter sampled February 4-13, 2015. The coarse filter Zn:Al ratio dropped to 0.02 during the
Saharan dust storm from the typical value of 0.3. On the coarse filter we obtained an absolute
CM mass of 950 μg, which accounts for half of the collected mass during the storm. 13% of dust
storm $PM_c$ is combined sea salt, $ANO_3$, and $ASO_4$, leaving 35% RM. Although this RM fraction
may imply an incomplete mass reconstruction, it is possible that a significant portion of desert
dust consists of organic material (Falkovich et al., 2004).
***4.12.9 Atlanta, US (n = 13)***
Atlanta represents a major urban area in a developed country. The temporary SPARTAN site
was located at the South Dekalb supersite 15 km east of downtown Atlanta. Air sampling was
performed for a 4-month period spanning winter to spring 2014. Over the past 10 years
significant decreases in $PM_{2.5}$ have been observed here and across the eastern United States
(Boys et al., 2014). The surrounding region is tree-covered or agricultural.
*Atlanta (South Dekalb) Collocation:* Multi-year averaged data from the Chemical Speciation
Network (CSN) Atlanta station (USEPA, 2015) provides recent data from 2007-2013 for
comparison with 2014 SPARTAN data. The EPA OM fraction (60%) agrees well with the
SPARTAN mean RM (49%). Crustal, $ANO_3$, EBC and $ASO_4$ are within 4% relative to total
composition. Aerosol component fractions in Atlanta are consistent with Butler et al. (2003).
Other components CM (10%), $ASO_4$ (21%), and $ANO_3$ (3%) closely match their values,
except for EBC (11% vs. 3%).
***4.12.11 Hanoi, Vietnam (n = 10)***
Hanoi is an inland megacity surrounded by grassland and agriculture. The sampler itself is on
a building rooftop at the Vietnam Academy of Science, 5 km northwest of the city center.
Motorbikes are the main forms of transportation downtown and the primary source of mobile-
based $PM_{2.5}$ (Vu Van et al., 2013). In Hanoi the $PM_{2.5}$ Zn:Al ratio was 2.6, indicative of





significant traffic and tire wear, is also the highest of any SPARTAN site. Biomass burning, coal
power, and cement are significant sources of $PM_{2.5}$ (Cohen et al., 2010).

*Hanoi Comparison:* SPARTAN differed with Cohen et al.(2010) regarding the contributions
of several compounds, perhaps related to differences in sampling season and location.
SPARTAN sea salt fraction was larger (2.5% vs. 0.6%), whereas $ASO_4$ (17%) was less than
reported by Cohen et al. (29%). Sulfate tends to be lower in the spring-summer seasons
(Cohen et al., 2010), coinciding with our measurement period, which may explain the
discrepancy. SPARTAN BC (9%) is close to Cohen et al. (2010), whereas SPARTAN RM
(49%) and CM (17%) masses are slightly larger.

**4.12.12 Singapore, Singapore (n = 6)**
Singapore is a densely populated coastal city-state at 7,770 people/$km^2$. The sampler is
located on a rooftop at the National University of Singapore (NUS), near the center of the city.
Transportation is mixed-use, including taxis, rail, and bicycles, which may help explain the
relatively low EBC and CM of 3%. Despite this, the Zn:Al ratio remains high at 1.6, implying a
dominant traffic-based contribution to CM. SPARTAN instruments have observed significant
biomass burning downwind from Indonesia, causing an increase in absolute $PM_{2.5}$ from 32 in
August to 120 $\mu g\ m^{-3}$ in September 2015, as well as an increase in RM from 44% to 62%. The
K:Al ratio steadily increased during this same period, from 7.2 (Jul 24 – Aug 2, 2015) to 17 – 24
(Aug 11 – Sept 25).

**4.12.12 Pretoria, South Africa (n = 5)**
Pretoria is a high-altitude city (1300 m) surrounded by arid, low-intensity agriculture and
extensive grasslands. The SPARTAN sampler is located on a 10m CSIR building rooftop 12 km
east of downtown area (*pop.* 700,000). Preliminary measurements of south-hemisphere
springtime show absolute $PM_{2.5}$ concentrations to be low, at 6.4 $\mu g\ m^{-3}$. There are significant
fractions of CM (22%) and EBC (22%), and low RM (14%). The Zn:Al ratio (0.69) indicates
vehicle traffic contributes to CM.
## 5. Refining estimates of dry hourly $PM_{2.5}$ using $\kappa_v$
Our assessment of $PM_{2.5}$ hygroscopicity is determined by site-specific chemical composition. We
then use the time-varying hygroscopicity to refine the $PM_{2.5}$ values inferred from nephelometer
scatter.
### 5.1. Relating $PM_{2.5}$ composition to $\kappa_v$
The outer pie charts of Figure 1 show the site-mean hygroscopic growth constant $\kappa_v$,
surrounded by the water contributions at 35% RH. The major contributors to PBW are $ASO_4$,
$ANO_3$, RM, and sea salt, as inferred from the values listed in Table 1 and weighted by
composition as in Eq. 5. $ASO_4$ and RM contribute similarly to total aerosol water whereas $ANO_3$
contributes less to $PM_{2.5}$ hygroscopicity due to its smaller mass. The contribution of sea salt to
hygroscopicity can be significant, and makes a dominant contribution in both Rehovot and
Buenos Aires.

The parameter $\kappa_v$, when averaged across all sites, is 0.19, similar to the generic estimate
$\kappa_{v,tot} = 0.2$ applied in the initial SPARTAN study (Snider et al., 2015). It is slightly larger than



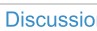
Atmospheric
Chemistry
and Physics
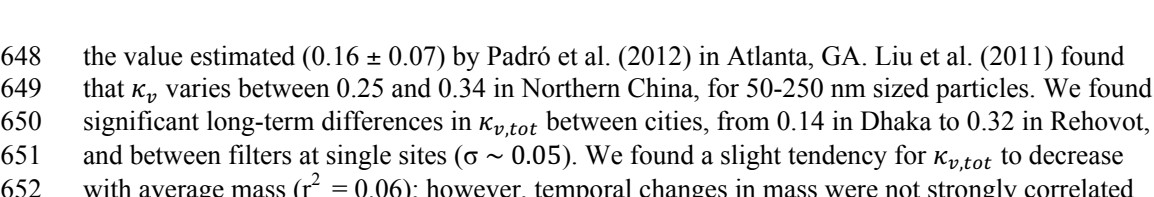
Discussions

the value estimated (0.16 ± 0.07) by Padró et al. (2012) in Atlanta, GA. Liu et al. (2011) found
that $\kappa_v$ varies between 0.25 and 0.34 in Northern China, for 50-250 nm sized particles. We found
significant long-term differences in $\kappa_{v,tot}$ between cities, from 0.14 in Dhaka to 0.32 in Rehovot,
and between filters at single sites (σ ~ 0.05). We found a slight tendency for $\kappa_{v,tot}$ to decrease
with average mass ($r^2 = 0.06$); however, temporal changes in mass were not strongly correlated
with $\kappa_{v,tot}$ ($r^2 < 0.01$). There are significant changes in $\kappa_{v,tot}$ due to seasonality and specific
events (e.g. dust storms, fires). In Beijing, aerosol hygroscopicity was 50% higher in mid
summer (August) due to increased sulfate, and in late winter (March) due to a relative increase in
sea salt. A summertime sulfate peak also agrees with observations by Yang et al. (2011).
Table 3 shows the site-specific PBW in $PM_{2.5}$. At RH =35%, PBW ranges from $1 - 6$ µg m$^{-3}$,
comparable to EBC. Above 80% RH, PBW will account for more than half of aerosol mass.
Accounting for this water component in nephelometer scatter motivates the following section.
**5.2. Relating nephelometer scatter to dry (RH=35%) $PM_{2.5}$**
We apply a temporally resolved, site-specific $\kappa_v$ to refine our relationship between total
nephelometer scatter and $PM_{2.5}$. We calculate a 45-day running mean aerosol volume-weighted
$\kappa_v$ at each SPARTAN site. We then use the hygroscopic growth factors to estimate dry hourly
$PM_{2.5}$ from hourly nephelometer measurements of ambient scatter and hourly measured RH.
Appendix A2 describes the procedure in more detail.
We compared our hourly $PM_{2.5}$ in Beijing with $PM_{2.5}$ measurements from a Beta Attenuation
Monitor (MetOne) at the US Embassy, located 15 km away. The left panel of Figure 4 shows the
time series of hourly dry $PM_{2.5}$ concentrations predicted by SPARTAN during the summer.
Pronounced temporal variation is apparent, with $PM_{2.5}$ concentrations varying by more than an
order of magnitude. A high degree of consistency is found with the BAM ($r^2 = 0.67$). The
exclusion of water uptake in hourly $PM_{2.5}$ estimates (by setting all $\kappa_v = 0$) decreased hourly
correlations slightly to $r^2 = 0.62$. The average humidity in Beijing was just 47% for the
measurement period, corresponding to a mean 17% volume contribution by water ($\kappa_v = 0.19$),
which approaches our measurement error (Appendix A2). Hygroscopic compensation should
play a more significant role under more humid conditions (e.g. Manila and Dhaka).
The right panel shows daily-averaged $PM_{2.5}$ ($n = 148$). In 2014 there were 3167
coincidentally available hours with which to compare. The coefficient of variation for averaged
24-hour measurements remained high ($r^2 = 0.71$). There was a mean offset of 10 µg m$^{-3}$.
However the slope is near unity (0.98), suggesting excellent proportionality between our
nephelometer and the BAM instrument for $PM_{2.5}$ concentrations below 200 µg m$^{-3}$. Above this
concentration, nephelometer signals become non-linear. The agreement remained similar for
hourly values ($r^2 = 0.67$).

## 6. Conclusions
We have established a multi-country network where continuous monitoring with a 3-
wavelength nephelometer is combined with a single multi-day composite filter sample to provide
information on $PM_{2.5}$. Long-term average aerosol composition is inferred from the filters,
including black carbon, sea salt, crustal material, ammonium sulfate, and ammonium nitrate.





This composition information was applied to calculate aerosol hygroscopicity, and in turn the
relation between aerosol scatter at ambient and controlled RH. These data provide a consistent
set of compositional measurements from 12 sites in 11 countries.
We report ongoing measurements of fine particulate matter ($PM_{2.5}$), including compositional
information, in 12 locations over a three-year span (2013-2015). The mean composition averaged
for all SPARTAN sites is 20% ammonium sulfate, 12% crustal material, 11% equivalent black
carbon, 4% ammonium nitrate, 7% particle bound water (at 35% RH), 2% sea salt, 1% trace
element oxides, and 43% residual matter.
Analysis of filter samples reveals that several $PM_{2.5}$ chemical components varied by more
than an order of magnitude between sites. Ammonium sulfate ranged from 1 µg m$^{-3}$ in Buenos
Aires to 17 µg m$^{-3}$ in Kanpur (dry season). Ammonium nitrate ranged from 0.2 µg m$^{-3}$
(Mammoth Cave, summertime) to 6.7 µg m$^{-3}$ (Kanpur, dry season). Equivalent black carbon
ranged from 0.7 µg m$^{-3}$ (Mammoth Cave) to 8 µg m$^{-3}$ (Dhaka, Bangladesh and Kanpur).
Crustal material concentrations ranged from 1 µg m$^{-3}$ (Atlanta) to 13 µg m$^{-3}$ (Beijing).
Measuring Zn:Al ratios in $PM_{2.5}$ was an effective way to determine anthropogenic contribution
to crustal material. Ratios larger than 0.5 identified sites with significant road dust contributions
(e.g. in Hanoi, Dhaka, Manila, and Kanpur). Some locations, such as Beijing and Buenos Aires,
had both high anthropogenic and natural crustal material. Low coarse Zn:Al ratios were apparent
during natural dust storms. Anthropogenic crustal material is an aerosol component neglected by
most global models and which may deserve more attention.
Potassium is a known marker for wood smoke; enhanced K:Al ratios were found in
Singapore downwind of Indonesian forest fires, in Kanpur during the winter dry season from
agricultural burning, and in Rehovot during a bonfire festival.
SPARTAN measurements generally agree well with previous collocated studies. SPARTAN
sulfate fractions are within 4% of fractions measured at eight of the ten collocated, though
temporally non-coincident, studies. Dedicated contemporaneous collocation with IMPROVE at
Mammoth Cave yielded a high degree of consistency with daily sulfate ($r^2 = 0.86$, slope = 1.03),
daily $PM_{2.5}$ ($r^2 = 0.76$, slope = 1.12), and mean fractions for all major $PM_{2.5}$ components (within
2%). Crustal material is typically consistent with the previous measurements, at 5-15%
composition. SPARTAN equivalent black carbon ranged broadly, from 3% (Singapore) to 25%
(Manila), and matched within a few percent of most previous works. Ammonium nitrate (4%)
generally matched other sites, though it was sometimes lower, as in Beijing and Atlanta. Sea-salt
was consistently less than 3% total mass, as found in previous measurements. Sea salt fractions
were highest in Buenos Aires and Rehovot (6%), reflecting natural coastal aerosols. SPARTAN
residual matter is consistent with the combined organic and unknown masses. Comparing with
collocated measurements supports the expectation that most of the RM is partially organic.
Residual matter could also include unaccounted-for particle bound water, measurement error,
and possibly unmeasured inorganic materials.
Seasonal tendencies are beginning to emerge within the SPARTAN study. Ammonium
sulfate concentrations remained relatively stable at 10-30% and reflect a variety of regional



combustion sources. In Kanpur, Beijing, and Dhaka, ammonium nitrate reached peak relative
concentrations (> 7 %) during wintertime, due to lower temperatures. By contrast, summertime
ammonium nitrate in Mammoth Cave was much lower, at 1.5%. Ammonium sulfate and residual
matter concentrations increase in tandem, which implies related sources. We also observed
relatively high crustal material mass fractions in Bandung during two months of 2014 volcanic
activity, and in the springtime for Beijing during regional dust episodes. Subsequent work will
explore temporal variation in detail.

We calculated the hygroscopic constant $\kappa_v$ for individual $PM_{2.5}$ filters to estimate water at
variable humidity, and to infer wet and water-free residual matter. Based on a range of literature,
we treated residual matter as mostly organic, with constant $\kappa_{v,RM} = 0.1$. Residual matter and
ammonium sulfate contributed the most to overall water uptake in aerosols. These individual
species, along with sea salt and ammonium nitrate, resulted in a mean mixed hygroscopic
constant of 0.19, implying that for many sites, water content above 80% RH will account for
more than half of aerosol mass. For cleanroom conditions of low humidity (35% RH), mean
water composition was estimated to be 7% by mass.

Water retention calculations allow for volumetric fluctuations estimates of aerosol water
at variable RH. We subtracted the water component to predict dry nephelometer scatter as a
function of time, anchored to filter masses at 35% RH. For Beijing, we assessed the consistency
of SPARTAN predictions of hourly $PM_{2.5}$ values with BAM measurements taken 15km away,
and found temporal consistency ($r^2 = 0.67$), with a slope near unity (0.98). The explained
variance decreased to $r^2 = 0.62$ when setting $\kappa_v = 0$. This comparison tested both SPARTAN
instrumentation and our treatment of aerosol water uptake.

These measurements provide chemical and physical data for future health research on
$PM_{2.5}$. Collocation with sun photometer measurements of AOD connects satellites observations
to ground-based measurements and provides information needed to evaluate chemical transport
model simulations of the $PM_{2.5}$ to AOD ratio. As sampling expands, SPARTAN will provide
long-term data on fine aerosol variability from around the world. Future work includes an
analysis of trace metal concentrations (Snider et al., in prep.) and applying SPARTAN
measurements to evaluate the $PM_{2.5}$ to AOD ratio in a chemical transport model (Weagle et al.,
in prep.). The data are freely available as a public good at www.spartan-network.org. We
welcome expressions of interest to join this grass-roots network.

**Acknowledgements**

The Natural Sciences and Engineering Research Council (NSERC) of Canada supported this
work. We are grateful to many who have offered helpful comments and advice on the creation of
this network including Jay Al-Saadi, Ross Anderson, Kalpana Balakrishnan, Len Barrie, Sundar
Christopher, Matthew Cooper, Jim Crawford, Doug Dockery, Jill Engel-Cox, Greg Evans,
Markus Fiebig, Allan Goldstein, Judy Guernsey, Ray Hoff, Rudy Husar, Mike Jerrett, Michaela
Kendall, Rich Kleidman, Petros Koutrakis, Glynis Lough, Doreen Neil, John Ogren, Norm
O'Neil, Jeff Pierce, Thomas Holzer-Popp, Ana Prados, Lorraine Remer, Sylvia Richardson, and
Frank Speizer. Data collection Rehovot was supported in part by the Environmental Health Fund
(Israel) and the Weizmann Institute. Partial support for the ITB site was under the grant HIBAH




WCU-ITB. The site at IIT Kanpur is supported in part by National Academy of Sciences and
USAID. The views expressed here are of authors and do not necessarily reflect those of NAS or
USAID.

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





**Figures and Tables**
**Table 1: κ-Kohler constants for volume ($\kappa_v$), mass ($\kappa_m$), and related quantities**

| Compound [X] | $\kappa_{v,X}$ | Approximate Density ($\rho_X/\rho_{water}$) | $\kappa_{m,X}$ | PBW(%mass) at RH = 35% | PBW(%mass) at RH = 80% |
|---|---|---|---|---|---|
| Crustal | 0 | 2.5[a] | 0 | 0 | 0 |
| EBC | 0 | 1.8[b] | 0 | 0 | 0 |
| TEO | 0 | 2.5 | 0 | 0 | 0 |
| RM | 0.1[c] | 1.4 | 0.07 | 2 | 12 |
| ANO$_3$ | 0.41 | 1.72 | 0.24 | 17 | 61 |
| ASO$_4$ | 0.51 | 1.76 | 0.29 | 15 | 56 |
| Na$_2$SO$_4$ | 0.68[d] | 2.68[d] | 0.25 | 12 | 50 |
| NaCl | 1.5[e] | 2.16 | 0.69 | 22 | 68 |

PBW = Particle-bound water. EBC = Equivalent black carbon, TEO = Trace Element Oxides, RM = Residue Matter
(associated with organics), ANO$_3$ = ammonium nitrate, ASO$_4$ = ammonium sulfate. [a] Wagner et al. (2009), [b]Bond
and Bergstrom (2006), [c]Assuming an urban O:C ratio of 0.5 then $\kappa_{v,OM} = 0.1$ Jimenez et al. (2009), [d]Petters and
Kreidenweis (2007). [e]Fitted using non-deliquesced, subsaturated AIM Model III values, plus 0% RH endpoint by
Kreidenweis et al. (2008).
**Table 2: Summary of speciation definitions**

| Species (at 0% RH) | Measurement Method | Species Mass (µg). For concentrations, divide masses by sampling volume $v$ | Reference |
|---|---|---|---|
| NaCl | IC (anion and cation) | $[Na^+]_{SS} = [Na^+]_{tot} - 0.1[Al]$ <br> $2.54[Na^+]_{SS}$ | (Remoundaki et al., 2013) (Malm et al., 1994) |
| ANO$_3$ | | $1.29[NO_3^-]$ | (Malm et al., 1994) |
| ASO$_4$ | | $[SO_4^{2-}]_{non-ss} + [NH_4^+] - 0.29[NO_3^-]$ where $[SO_4^{2-}]_{non-ss} = [SO_4^{2-}]_{total} - 0.12[Na^+]$ | (Dabek-Zlotorzynska et al., 2011; Henning et al., 2003) |
| Na$_2$SO$_4$ | | $0.18[Na^+]_{SS}$ | |
| CM | ICP-MS & IC | $([Al] + [Mg] + [Fe])/0.1$ | (Wang, 2015) |
| EBC | SSR | $20.7 \times \ln(R_o/R)$ | (Taha et al., 2007) |
| TEO | ICP-MS | $1.47[V] + 1.27[Ni] + 1.25[Cu] + 1.24[Zn] + 1.32[As] + 1.2[Se] + 1.07[Ag] + 1.14[Cd] + 1.2[Sb] + 1.12[Ba] + 1.23[Ce] + 1.08[Pb]$ | (Malm et al., 1994) |
| PBW$_{inorg}$ | $\kappa_{m,X}$ | $\sum_X [f_{m,X}(RH) - 1][X]$ | (Kreidenweis et al., 2008) |
| PBW$_{RM}$ | | $RM(1 - 1/f_{m,RM})$ | Table 1 |
| RM(35%) | Mass Balance | $[PM_{2.5}] - \{[EBC] + [CM] + [TEO] + [ANO_3] + [NaCl] + [ASO_4] + [Na_2SO_4] + [PBW_{inorg}]\}$ | This Study |
| RM(0%) | Mass Balance $\kappa_{m,OM} = 0.07$ | $RM(35\%) - PBW_{RM}$ | Organic growth factors: (Jimenez et al., 2009; Sun et al., 2011) |

**Species**: EBC = Equivalent black carbon, TEO = Trace metal oxides, CM = Crustal Material, ANO$_3$ = Ammonium
nitrate, ASO$_4$ = Ammonium sulfate, PBW = particle-bound water, RM = residue matter (assumed representative of
organic matter). **Measurement Instruments:** IC = Ion Chromatography, ICP-MS Inductively coupled plasma mass
spectrometry, $\kappa_{m,X}$ = single-parameter hygroscopicity by mass (Kreidenweis et al., 2008).



**Table 3: PM$_{2.5}$ composition and water content (µg m$^{-3}$) at each SPARTAN location.**

| City | Host Institute | Lat/Lon (°) | Elev.//Inst. Elev. (m) | Filters (n) | ASO$_4$ | ANO$_3$ | CM | NaCl | EBC | TEO | RM | PBW 35%RH | ρ 0%RH (g/cm³) | PM$_{2.5}$ | PM$_{2.5}$/PM$_{10}$ | κ$_{v,tot}$ | PM$_{2.5}$ Zn/Al | Filter Sampling Period |
|---|---|---|---|---|---|---|---|---|---|---|---|---|---|---|---|---|---|---|
| Beijing | Tsinghua University | 40.010, 116.333 | 60// 7.5 | 100 | 12.0 (7.6)[a] | 4.7 (5.8) | 13.4 (7.1) | 1.3 (2.1) | 5.6 (3.6) | 0.62 (0.32) | 28.4 (19.5) | 4.7 (2.4) | 1.64 | 67.9 (2.6) | 0.48 | 0.19 | 0.45 | 2013/06 – 2015/10 |
| Bandung | ITB Bandung | -6.888, 107.610 | 826// 20 | 71 | 6.0 (2.3) | 0.5 (0.4) | 2.4 (1.2) | 0.3 (0.1) | 3.9 (2.0) | 0.17 (0.10) | 15.5 (5.6) | 1.9 (0.6) | 1.56 | 30.6 (1.0) | 0.58 | 0.17 | 0.53 | 2014/01 – 2015/09 |
| Manila | Manila Observatory | 14.635, 121.080 | 60// 10 | 56 | 2.7 (1.5) | 0.3 (0.2) | 1.8 (1.2) | 0.4 (0.2) | 4.3 (3.3) | 0.12 (0.10) | 7.4 (3.6) | 1.0 (0.4) | 1.61 | 17.9 (0.9) | 0.39 | 0.15 | 0.89 | 2014/02 – 2015/09 |
| Dhaka | Dhaka University | 23.728, 90.398 | 20// 20 | 41 | 7.5 (4.3) | 2.1 (1.8) | 6.7 (3.2) | 1.4 (1.7) | 8.0 (6.2) | 1.50 (1.46) | 21.1 (14.9) | 3.5 (2.1) | 1.64 | 51.9 (3.7) | 0.4 | 0.17 | 2.68 | 2014/05 – 2015/11 |
| Ilorin | Ilorin University | 8.484, 4.675 | 330// 10 | 40 | 1.9 (0.8) | 0.3 (0.1) | 2.8 (1.8) | 0.3 (0.4) | 1.6 (0.8) | 0.11 (0.07) | 7.8 (3.9) | 0.9 (0.4) | 1.61 | 15.7 (0.8) | 0.44 | 0.15 | 0.53 | 2014/03 – 2015/10 |
| Kanpur | IIT Kanpur | 26.519, 80.233 | 130// 10 | 33 | 17.3 (11.8) | 6.7 (5.3) | 4.2 (3.0) | 0.6 (0.3) | 8.1 (4.7) | 0.62 (0.47) | 53.3 (33.9) | 6.2 (3.6) | 1.52 | 97.2 (9.2) | 0.56 | 0.18 | 1.36 | 2013/12 – 2014/11 |
| Buenos Aires | CITEDEF | -34.560, -58.506 | 25// 7 | 21 | 1.1 (0.3) | 0.8 (0.7) | 2.2 (0.8) | 0.6 (0.3) | 1.1 (1.3) | 0.12 (0.12) | 2.7 (2.3) | 0.9 (0.3) | 1.72 | 10.0 (0.6) | 0.37 | 0.20 | 0.42 | 2014/10 – 2015/04 |
| Mammoth Cave NP | Mammoth Cave | 37.132, -86.148 | 235// 7 | 19 | 4.1 (2.4) | 0.2 (0.1) | 1.4 (1.3) | 0.1 (0.1) | 0.7 (0.4) | 0.03 (0.02) | 6.3 (4.5) | 1.0 (0.6) | 1.58 | 13.6 (1.8) | 0.56 | 0.22 | 0.13 | 2014/06 – 2014/08 |
| Rehovot | Weizmann Institute | 31.907, 34.810 | 20// 10 | 19 | 4.8 (1.6) | 0.7 (0.4) | 2.1 (0.5) | 0.9 (0.6) | 2.0 (2.4) | 0.09 (0.05) | 2.7 (3.0) | 1.7 (0.7) | 1.73 | 15.2 (1.3) | 0.41 | 0.31 | 0.59 | 2015/02 – 2015/08 |
| Atlanta | Emory University | 33.688, -84.290 | 250// 2 | 13 | 2.0 (0.9) | 0.3 (0.1) | 0.9 (0.4) | 0.1 (0.1) | 0.9 (1.0) | 0.04 (0.02) | 4.1 (1.8) | 0.6 (0.2) | 1.63 | 9.1 (0.7) | 0.69 | 0.17 | 0.31 | 2014/01 – 2014/05 |
| Hanoi | Vietnam Acad. Sci. | 21.048, 105.800 | 10// 20 | 10 | 6.0 (2.1) | 1.6 (0.4) | 6.1 (4.6) | 0.9 (0.2) | 3.7 (2.1) | 0.70 (0.37) | 17.8 (7.7) | 2.6 (0.7) | 1.59 | 39.4 (3.9) | 0.38 | 0.18 | 2.57 | 2015/05 – 2015/08 |
| Singapore | NUS | 1.298, 103.780 | 10// 20 | 6 | 17.6 (7.3) | 1.5 (1.2) | 2.4 (0.5) | 1.1 (0.4) | 2.4 (1.4) | 0.22 (0.06) | 39.8 (27.2) | 5.5 (2.6) | 1.51 | 70.6 (16.2) | 0.77 | 0.21 | 1.57 | 2015/08 – 2015/09 |
| Pretoria | CSIR | -25.756, 28.280 | 1310// 10 | 5 | 1.2 (1.6) | 0.7 (0.3) | 1.4 (1.6) | 0.2 (0.1) | 1.4 (0.9) | 0.07 (0.04) | 0.9 (0.7) | 0.5 (0.4) | 1.97 | 6.4 (2.3) | 0.32 | 0.29 | 0.69 | 2015/09- 2015/11 |
| SPARTAN mean (% mass) | All sites | | | 437 | 20 (10)% | 4.0 (2.8)% | 11.9 (6.2)% | 2.2 (1.5)% | 10.7 (8.4)% | 0.85 (0.63)% | 43 (25)% | 7.2 (3.1)% | 1.61 | 35.2 (3.6) | 0.49 | 0.194 | 0.69 | **2013 – 2015** |

[a]Values in parentheses are 1σ standard deviations. ANO$_3$ = ammonium nitrate, ASO$_4$ = ammonium sulfate, CM = Crustal material, EBC = Equivalent black carbon, TEO = Trace Element Oxides, RM = Residue Matter, PBW = Particle-bound water. Mean Na$_2$SO$_4$ was not significant (< 0.1 µg m$^{-3}$) at any SPARTAN site. [b]Geometric mean of ratio



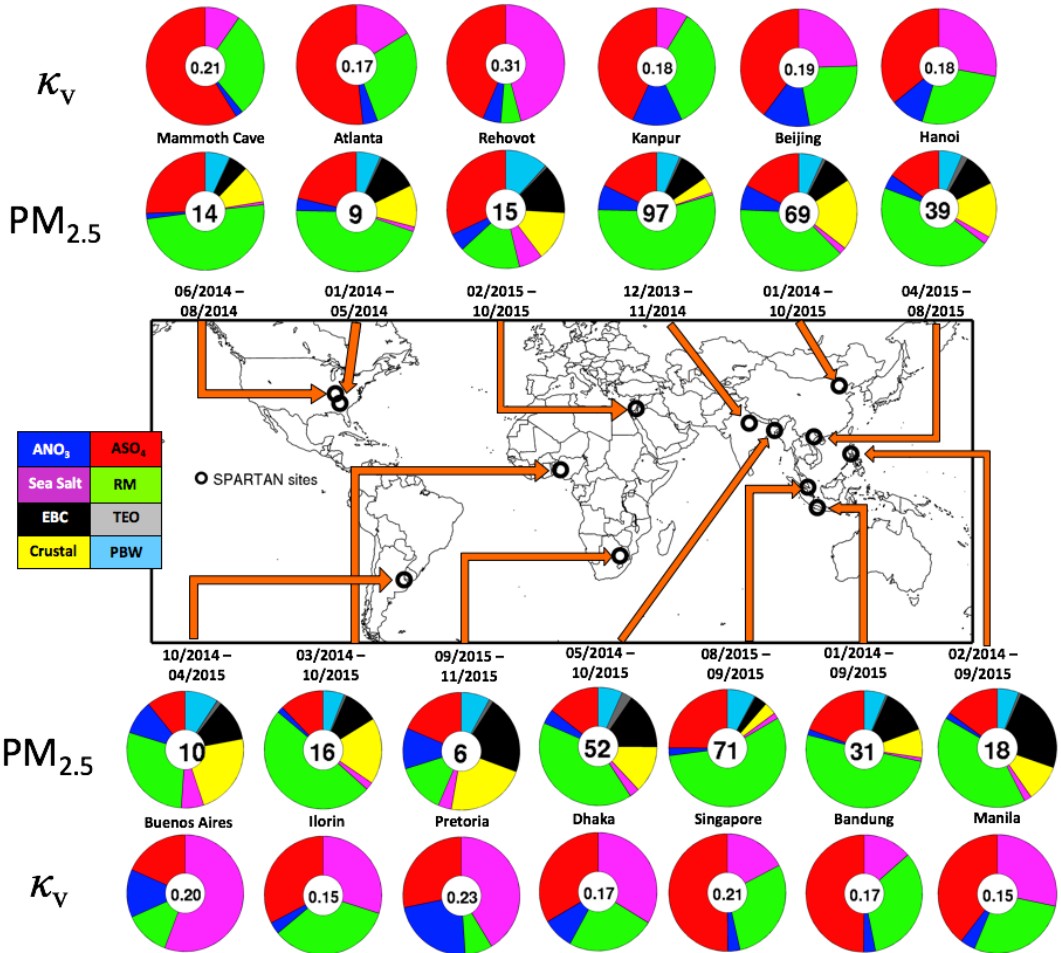

**Figure 1: PM$_{2.5}$ mass (inner circle, μg m$^{-3}$) and composition mass fraction (filled colors) is shown in interior pie charts. Exterior pie charts contain site-mean $\kappa_v$ surrounded by the relative contribution of PBW water at 35% RH.**

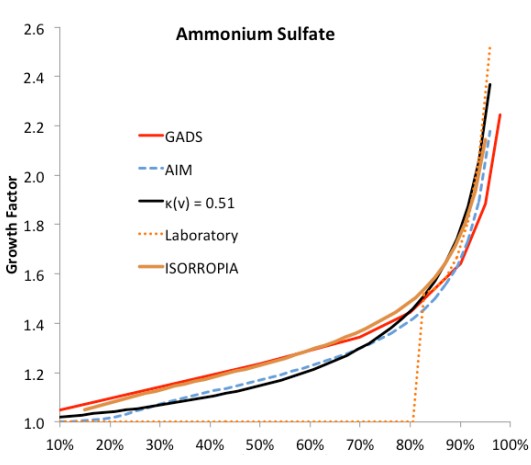

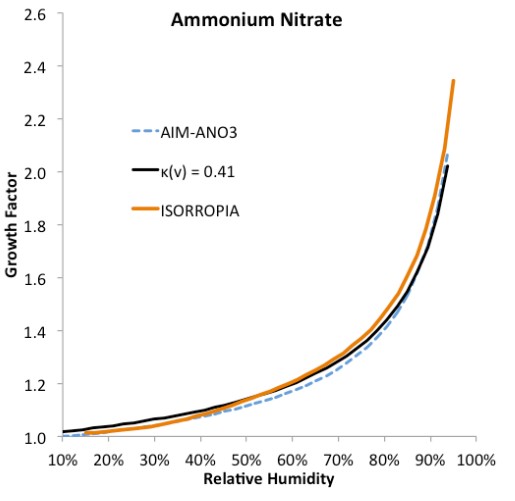

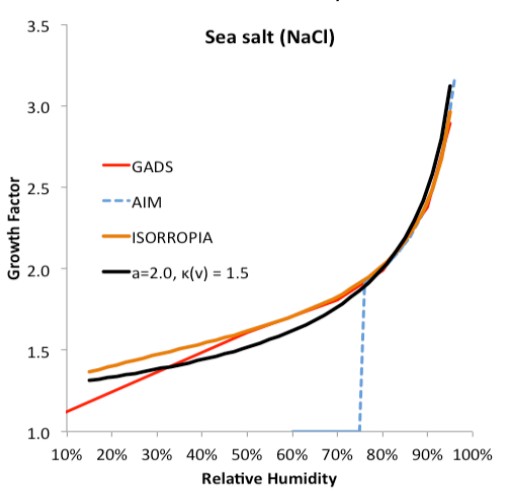

**Figure 2: Hygroscopic growth factors for ASO$_4$ (left), ANO$_3$ (centre), and sea salt (right). GADS = Global Aerosol Dataset estimated from empirical data** (Koepke et al., 1997)**. ISORROPIA = Aerosol thermodynamic model at T=298K (reverse mode) and assuming linear water/solvent volume additivity** (Fountoukis and Nenes, 2007)**. AIM = Aerosol Inorganic Model calculated metastable growth for ASO$_4$ and ANO$_3$ at T=298K** (Wexler and Clegg, 2002)**, Laboratory ASO$_4$ fit is $GF = 1.49 + 2.81 \cdot RH^{24.6}$ (with deliquescence at 80%) is for bulk pure ASO$_4$** (Wise et al., 2003)**. All components are fit using Eq 3.**



**Figure 3: Comparison of SPARTAN water-free aerosol composition with 11 collocated speciation studies. The numbers in parentheses show 1-σ deviations of averaged masses. The number of filters sampled is _n_. Dark green = organic, Light green = residue, black = equivalent black carbon, red = ammonium sulfate, blue = ammonium nitrate, purple = sea salt, yellow = crustal, and grey stripes = unknown. OM/OC ratios are from Philip et al. (2014b) and Canagaratna et al. (2015). Relative mass percentages are based on water-free aerosol components. SPARTAN percentages do not sum to 100% due to omission of species not found in comparison studies.**

| PM$_{2.5}$ mass = μg m$^{-3}$ (1σ/√n), components = % (1σ) | | |
|---|---|---|
| **This study (total mass = μg m$^{-3}$)** | **Study A (μg m$^{-3}$)** | **Study B (μg m$^{-3}$)** |
| **Beijing** PM$_{2.5}$: 70 (3), _n_ = 100<br>7 (9)% ANO$_3$,<br>19 (12)% ASO$_4$,<br>2.1 (3.5)% NaCl,<br>21 (11)% CM,<br>9.0 (3.5)% EBC,<br>42 (30)% RM | (Yang et al., 2011) 2005-2006,<br>OM/OC = 1.7,    PM$_{2.5}$: 119(40)<br>11 (7)% ANO$_3$,<br>17 (10)% ASO$_4$,<br>1.3 (0.6)% NaCl,<br>19 (3)% CM,<br>7 (5)% EC,<br>33 (16)% OM,<br>10 (10)% Unk | (Oanh et al., 2006) 2001-2004,<br>OM/OC = 1.7  PM$_{2.5}$: 136 (45)<br>12 (1.5)% ANO$_3$,<br>20 (1.8)% ASO$_4$,<br>1.2 (1.2)% NaCl,<br>5 (3)% CM,<br>9 (7)% EBC,<br>29 (22)% OM,<br>24 (24)% Unk |
| **Bandung** PM$_{2.5}$: 31 (1), _n_ = 71<br>1.8 (1.4)% ANO$_3$,<br>20 (8)% ASO$_4$,<br>1.1 (0.4)% NaCl,<br>8.4 (4.2)% CM,<br>14 (4)% EBC,<br>54 (20)% RM | (Oanh et al., 2006) 2001-2004.<br>OM/OC = 2.2,   PM$_{2.5}$: 45.5(10.6),<br>13(4)% ANO$_3$,<br>21(3)% ASO$_4$,<br>1.6(0.2)% NaCl,<br>6.6(0.5)% CM,<br>19 (4)% EBC,<br>36(11)% RM | (Lestari and Mauliadi, 2009) 2001- 2007, OM/OC<br>= 2.2  PM$_{2.5}$: 43.5(10.5)<br>4(6)% ANO$_3$,<br>4(4)% ASO$_4$,<br>3(2)% NaCl,<br>23(21)% CM,<br>24(14)% EBC,<br>42(35)% RM |
| **Manila** PM$_{2.5}$: 18 (1), _n_ = 56<br>1.8 (1.2)% ANO$_3$,<br>16 (9)% ASO$_4$,<br>2.4 (1.2)% NaCl,<br>11 (3)% CM,<br>25 (12)% EBC,<br>44 (22)% RM | (Cohen et al., 2009) 2001-2007,<br>OM/OC = 2.1,    PM$_{2.5}$: 46 (19),<br>ANO$_3$ N/A<br>14 (9)% ASO$_4$,<br>0.6 (1.5)% NaCl,<br>5 (1.7)% CM,<br>25 (11)% EBC,<br>57(22)% OM, | |
| **Kanpur** PM$_{2.5}$: 97 (9), _n_ = 33<br>7.4 (6.7)% ANO$_3$,<br>19 (15)% ASO$_4$,<br>0.7 (0.3)% NaCl,<br>4.7 (2.9)% CM,<br>9 (5.0)% EBC,<br>59 (50)% RM | (Behera and Sharma, 2010) Oct. 2007 – Jan 2008,<br>OM/OC = 2.2,   PM$_{2.5}$: 172 (73),<br>6.1 (1.3)% ANO$_3$,<br>18 (4)% ASO$_4$,<br>2.6 (0.6)% NaCl,<br>10 (3)% CM,<br>4.8 (1.1)% EC,<br>42 (9)% OM,<br>16 (10)% Unk | (Ram et al., 2012) Dec 2008 – Feb 2009,<br>OM/OC = 2.2  PM$_{2.5}$: 158 (47)<br>6.6(4)% ANO$_3$,<br>13 (5)% ASO$_4$,<br>1.5 (0.9)% NaCl,<br>12 (6)% CM*,<br>3 (1.1)% EC,<br>57 (23)% OM,<br>6 (24)% Unk<br>*Assuming CM = [Ca]/0.034 (Wang, 2015) |
| **Mammoth Cave NP** PM$_{2.5}$: 13.6 (2), _n_ = 19<br>1.1 (0.9)% ANO$_3$,<br>30 (14)% ASO$_4$,<br>0.6 (0.8)% NaCl,<br>10 (10)% CM,<br>4.0 (2.5)% EBC,<br>47 (40)% RM | (IMPROVE, 2015) June-Aug. 2014,<br>OM/OC = 2.0,   PM$_{2.5}$: 10.0 (5.8),<br>2.4 (2.5)% ANO$_3$,<br>36 (17)% ASO$_4$,<br>0.3 (1.6)% NaCl,<br>7 (8)% CM,<br>3 (3)% EC,<br>34 (30)% OM,<br>17% Unk+H$_2$O | |
| **Atlanta** PM$_{2.5}$: 9.1 (1), _n_ = 13<br>3.6 (1.3)% ANO$_3$,<br>24 (12)% ASO$_4$,<br>1.2 (1.3)% NaCl,<br>11 (6.6)% CM,<br>11 (2.6)% EBC,<br>49 (25)% RM | (Butler et al., 2003) Mar. 1999 –2000 Feb,<br>OM/OC = 2.0, PM$_{2.5}$: 24.2<br>4 (0.2)% ANO$_3$,<br>28 (1.0)% ASO$_4$,<br>10 (0.8)% CM,<br>3 (0.2)% EC,<br>55 (5)% OM, | EPA 2007 –2013 (USEPA, 2015), OM/OC = 2.0<br>PM$_{2.5}$: 15.3<br>5.0 (5)% ANO$_3$,<br>21 (15)% ASO$_4$,<br>0.6 (0.6)% NaCl<br>7.3 (5)% CM,<br>7.2 (5)% EC,<br>60 (36)% OM, |
| **Hanoi** PM$_{2.5}$: 39 (4), _n_ = 10<br>4.4 (1.1)% ANO$_3$,<br>17 (6)% ASO$_4$,<br>2.5 (0.6)% NaCl,<br>17 (13)% CM,<br>10 (3.3)% EBC,<br>49 (21)% RM | (Cohen et al., 2010). 2001 –2008<br>OM/OC = 2.1,  PM$_{2.5}$: 54 (33)<br>ANO$_3$ N/A<br>29 (20)% ASO$_4$,<br>0.6 (1.4)%NaCl<br>13 (7)% CM,<br>8 (3)% EBC,<br>40 (19)% OM,<br>2 (2)% Unk + ANO$_3$ | |





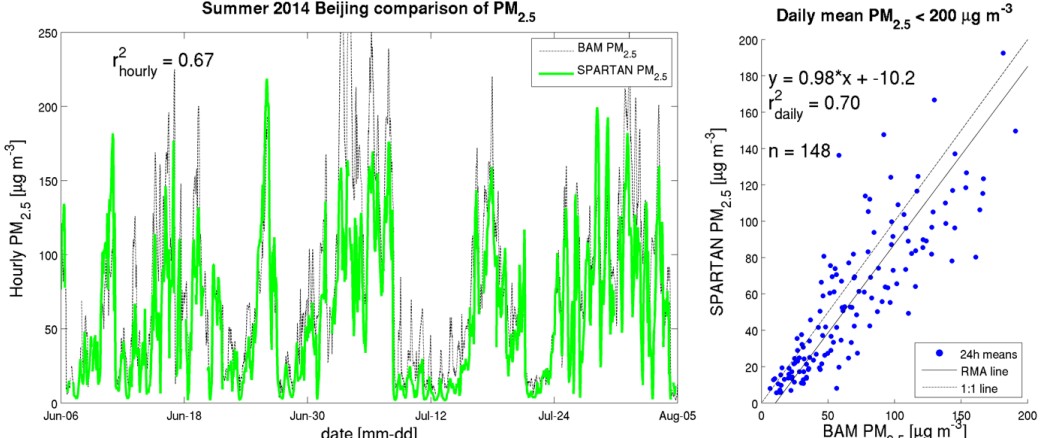

**Figure 4: Left Hourly PM$_{2.5}$ estimated from SPARTAN overlaid with a MetOne BAM-1020 (June-August 2014) at the Beijing US Embassy (15 km away). Right: 24-hour PM$_{2.5}$ predictions compared with BAM for the year 2014. Reduced major axis (RMA) slope and Pearson correlations for PM$_{2.5}$ are given in inset.**





# Appendix:

**Appendix A1**:

**Table A1: Hygroscopicity parameter $\kappa_v$ for various studies on organic material**

| $\kappa_v$ (OM) | Comments | Reference |
|---|---|---|
| 0.045 | Fitted to an aged organic mixture, subsaturated | (Varutbangkul et al., 2006) |
| 0 | IMPROVE network, subsaturated | (Hand and Malm, 2006) |
| 0.10 ± 0.04 | RH > 99%, fitted to SOA precursors | (Prenni et al., 2007) |
| $-0.067 + 0.33$(O:C) | Fitted, RH > 99% | (Jimenez et al., 2009) |
| $0.29$(O:C) | RH > 99%, 0.3 < O:C < 0.6 | (Chang et al., 2010) |
| 0.05 | Best estimate from aged mixtures, subsaturated | (Dusek et al., 2011) |
| 0.01 – 0.2 | Field studies & smog chamber, subsaturated | (Duplissy et al., 2011) |
| 0.16 | RH > 99% | (Asa-Awuku et al., 2011) |
| 0.05 – 0.13 | Lab experiments, aged with $H_2O_2$ and light; subsaturated | (Liu et al., 2012) |
| 0.1 | RH > 99%, $D_{dry}$ < 100 nm | (Padró et al., 2012) |
| $0.12\varepsilon_{WSOM}$[#] | RH > 99% | (Lathem et al., 2013) |
| $-0.005 + 0.19$(O:C) | Fitted, RH > 99% 100 nm particle | (Rickards et al., 2013) |
| 0.03, 0.1 | HDTMA-measure, subsaturated | (Bezantakos et al., 2013) |
| 0.1 | Subsaturated | Selected for this study |

[#]$\varepsilon_{WSOM}$ = fraction of water-soluble organic material.



**Appendix A2:**

Dry aerosol scatter ($b_{sp,\text{dry}}$) is related to relative humidity (RH) by

$$b_{sp,\text{dry}} = \frac{b_{sp}(\text{RH})}{f_v(\text{RH})} \qquad \text{Eq. A1}$$

Changes in scatter are also proportional to mass (Chow et al., 2006; Wang et al., 2010)

$$b_{sp,\text{dry}} = \alpha \text{PM}_{2.5,\text{dry}} \qquad \text{Eq. A2}$$

where $\alpha$ (m$^2$ g$^{-1}$) is the mass scattering efficiency and a function of aerosol size distribution, effective radius, and dry composition. In this study we treat composition, density, and size distribution as constant over each of our 9-day intermittent sampling periods so that $\alpha \approx \langle\alpha\rangle_{9d}$. Under this assumption the predicted mass changes in low humidity (35% RH) are proportional to water-free aerosol scatter:

$$\text{PM}_{2.5,\text{dry}} = \langle\text{PM}_{2.5,\text{dry}}\rangle \frac{b_{sp,\text{dry}}}{\langle b_{sp,\text{dry}}\rangle} \qquad \text{Eq. A3}$$

where < > indicates 9-day averages. The explicit compensation for aerosol water is then

$$[\text{PM}_{2.5,\text{dry}}] = \frac{\langle[\text{PM}_{2.5,\text{dry}}]\rangle}{\langle b_{sp}(\text{RH})/f_v(\text{RH})\rangle} \cdot \frac{b_{sp}(\text{RH})}{f_v(\text{RH})} \qquad \text{Eq. A4}$$

where [] indicates concentration in μg m$^{-3}$. Uncertainties are a function of replicate weighing measurements (± 2 μg), flow volume (± 10%), %RH (± 2.5), aerosol scatter (± 5%), and κ$_v$ (± 0.05).

$$\left(\frac{\delta[\text{PM}_{2.5,h}]}{[\text{PM}_{2.5,h}]}\right)^2 \approx \left(\frac{\delta\text{PM}_{2.5}}{\text{PM}_{2.5}}\right)^2 + \left(\frac{\delta V}{V}\right)^2 + \left(\frac{\delta b_{sp}}{b_{sp}}\right)^2 + \left(\frac{\delta f_v}{f_v}\right)^2 \qquad \text{Eq. A5}$$

where

$$\left(\frac{\delta f_v}{f_v}\right)^2 = \frac{(f_v-1)^2}{f_v^2}\left[\left(\frac{\delta\kappa}{\kappa}\right)^2 + \left(\frac{\delta\text{RH}}{\text{RH}\cdot(100-\text{RH})}\right)^2\right] \qquad \text{Eq. A6}$$

The average relative 2-σ PM$_{2.5}$ uncertainty was 26% for dry hourly predictions, increasing with higher RH cutoffs. A cut-off of RH = 80% has been applied to our data, above which hygroscopic uncertainties, as well as total water mass, dominate.