# Peer review of "Variation in Global Chemical Composition of PM2.5: Emerging Results from SPARTAN"

_Atmospheric Chemistry and Physics, 2016_

## Referee Comment (RC1) · Anonymous Referee #2 · 16 Mar 2016

Dear Editor,

This MS presents the results from the chemical characterisation of PM2.5 from 12 stations across the globe in the framework of the SPARTAN project. Nephelometry data are also presented, as well as an assessment of the influence of hygroscopicity. The study is well designed and interesting mainly due to its global scope, although precisely because of this global scope the results seem at times too general and it is difficult to understand the ultimate objective of the authors' work. Another issue is that limitations of the work are barely presented or discussed, especially obvious ones such as the absolute lack of data representation in Europe. This should be discussed. Also, the temporal representativeness of the data should be discussed given that the sampling period varied largely between sites (2 to 22 months) and clear outliers were present (dust storms, forest fires, etc.) Overall I would recommend publication after the

changes below are addressed, and especially after the overall aim of the manuscript (otherwise the MS as it is is merely descriptive) and the issue of temporal representativeness are discussed.

Specific comments:

- line 22: "2-22 month periods", what was the average period for all sites? Whether 2 or 22 months of data were available for a given station is highly relevant with regard to its representativeness. Please clarify in the abstract and in the text.

- line 34: the term "residue" is rather confusing, especially if one sees later on in the text that it refers mainly to organic matter. I would suggest to find a different term.

- line 51, treatment of outliers: how were outliers processed? This line refers to a major sulphate event, and in the final section of the MS for several stations the authors describe the impact of dust events, major forest fires... How were these outliers dealt with? Depending on the duration of the sampling (see the first comment above), these outliers may have been not representative and strongly impact the mean aerosol concentration. A section should be added to discuss how frequent these outliers were, and how they were treated.

- lines 116-121: please highlight as a limitation that no data are available for Europe. This is a major spatial gap, despite data being openly available (e.g., chemical speciation data and nephelometry from the EMEP stations, stored in the EBAS database).

- line 135, what was the diameter of the filters, 47 mm?

- line 155: were all the filters from all sites shipped to Dalhousie Univ. for analysis? Please describe how filters were stored + transported to guarantee sample conservation, as this could be a major issue. How many filters/month were sampled, and how many in total per site? Please add a Table (even if in Supplementary material).

- line 298: please add a reference for the 0.18 coefficient applied to Na+

- line 320: please rename the fraction "residue matter" and state clearly from the beginning that it refers mainly to organics

- section 4.10: this entire section is very descriptive, but hardly any interpretation of the results is provided.

- line 357: this result is surprising for Beijing: even despite the influence of dust storms, a major urban area like Beijing should have a high Zn/Al ratio (high anthrop. influence). Whys is this not the case? Could it be due to the potential lack of representativeness of the samples, linked to how outliers were dealt with? Please discuss and clarify in the text.

- line 376, 60% in Kanpur: if RM is so high in Kanpur it implies that CM will be very low, which seems surprising for an Indian city. Could this be because sampling took place during the monsoon season, maybe? This refers again to the temporal representativeness of the sampling. Some of the chemical composition results seem unexpected, please include a section discussing the potential sources of uncertainty, e.g. sampling period, filter transport, technical issues during sampling... What could have gone wrong, with such a disperse network of stations?

- line 394, strong correlation between ASo4 and ANO3: this is unexpected due to the thermal instability of ANO3 in summer (when formation of ASO4 is highest). On an annual basis both components might correlate, but not on a monthly basis.Please clarify.

- section 4.11: this section is very unclear, what studies are the authors comparing their SPARTAN results with? Please specify. What are "study A" and "Study B" in the Figure?

- line 417: "expectation that RM is organic", this is not an expectation, it is a definition of the SPARTAN methodology (given that all other components are already accounted for)

[Figure]

- line 410: "10-30%", this is an overall limitation of the work: because such a broad range of station types, geographical regions and locations is used, the results also become rather broad and non-specific. E.g., stating that ASO4 contributes with 10-30% to PM2.5 mass is not a very specific result, it could represent almost anywhere in the world. Therefore, please state this limitation in the intro or results section, and extract the use of this kind of very global data, e.g. probably (in my opinion) for modelling studies, etc. What is the ultimate purpose of this work? Otherwise it becomes simply a descriptive manuscript.

- section 4.12.1: same issue as above, please discuss the relative contribution of natural and anthropogenic dust in Beijing. The large number of samples available (100) should allow for this kind of interpretation.

- line 445: due to volatilzation but also to the fact that the authors are comparing their results with those from different periods in time, in the other studies.

- line 520, "organic matter burning": this is another example of a potential outlier. In addition, it would be useful if the authors could add somewhere in the text a brief assessment of major common emission sources, e.g., biomass combustion, agriculture, natural dust... This would help to integrate the results from the different sites rather than simply state ranges of chemical components (e.g., ASO4 = 10-30%) which don't provide much specific information.

- line 572: festival in Israel, another probable outlier

- line 691: black carbon should be equivalent black carbon

- line 697: "3-year span", please clarify that this is not a continuous monitoring period in all sites, but instead a sequence of consecutive 4-month (approx) periods in different stations. This is a very big difference.

- Conclusions: the conclusions section is again very descriptive, it is rather a summary. Actual conclusions and applications for their data should be extracted by the authors.

As proposed above, a reference to major common emission sources could be added (this is slightly hinted at in paragraph 7369-744).

---

## Referee Comment (RC2) · Anonymous Referee #3 · 17 Mar 2016

Review of "Variation in Global Chemical Composition of PM2.5: Emerging Results from SPARTAN" by Snider et al.

This paper summarizes measurements of aerosol composition, estimates of hygroscopicity, and particle bound water at 12 sites across the globe. The results suggest that on average across all sites PM2.5 mass constituents were (highest to lowest): residual material (assumed to be organics), ammoniated sulfate, crustal material, equivalent black carbon, particle bound water, ammonium nitrate, sea salt, and trace element oxides. The results represent considerable effort and a significant contribution to understanding particulate matter constituents at urban sites in different environments. The authors have done a commendable job of summarizing and presenting consistent measurements, and for making these data available to the community. The work is scientifically sound and for the most part the methods are well-described. The impor-

tance of water biases in filter sampling can be significant and a strength of the paper is the estimates of particle bound water and hygroscopicity based on chemical composition and assumptions of molecular speciation and hygroscopic properties. Strengths of the paper include the careful, consistent approach to summarizing data from many different sites, and interpreting the data in the context local and regional sources, including comparisons to other studies when available. The paper could benefit from some reorganization to help with clarity, and careful reading and editing to account for discrepancies between values stated in the text and reported in figures. Some discussion of the values of the average mass scattering efficiencies used for converting light scattering to mass should be included, and whether these were appropriate for varying composition.

Organizational Comments

The introduction is a bit disjointed. As a suggestion, I recommend moving the paragraph that begins on line 102 to line 78 (becomes 2nd paragraph), followed by the paragraph starting on line 93, with the sentence starting as "Furthermore, no global..." on line 80-82 added to the end, so that the RH effects are included in 1 paragraph. The rest of the paragraph (line 83-91) can become the 4th paragraph, followed by the paragraph at line 114, finishing with the paragraph on line 121. The order of the discussion would then be 1) Health (2) Chemical composition (3) Humidity effects (4) Satellites (5) SPARTAN (6) Purpose

A difficulty with the current organization of the paper is that the section on hygroscopicity comes before aerosol composition such that aerosol components (like ASO4) are being discussed before the reader knows how the authors define them. It would help to follow the development of the method if the description of the assumed aerosol components came first, and the kappa development followed. I recommend switching the order of section 3 (aerosol hygroscopicity) and section (4) PM2.5 aerosol composition. Also, define a new section for mass speciation results (4.10 and onward). In accord with this reorganizing, switch the order of Figure 1 and 2, and Tables 1 and 2.

Specific comments follow:

I recommend instead of "ammonium sulfate", referring to ASO4 as "ammoniated sulfate" because the definition doesn't necessarily assume fully neutralized ammonium sulfate (1.375*SO4).

Please point out that PM2.5 and PM10 are gravimetrically weighed.

Line 19: I'm not sure what "maximize the chemical and physical information" means? It seems like the project is characterizing the chemical and physical attributes of aerosols from filter samples.

Line 24: Define AERONET for first time use.

Line 28: Consider replacing "baseline" with "background" or "rural/remote".

Line 34: What RH?

Line 41: Define IMPROVE for first time use.

Line 42: From the slope, which network had higher mass?

Line 51: Change "included" to "including"

Line 51: What was the standard deviation?

Line 96: Define CSN

Line 97: Define AIM

Line 99: Define kappa

Line 126: "As a function of chemical speciation" seems redundant.

Line 134: Provide years of sampling

Line 147: Define PM10

Line 157: Is PM2.5 here gravimetric mass or summed constituents?

Line 166: Add "with other networks"

Line 173: This sentence implies that surface reflectance is used to obtain all of the following constituents, not just black carbon.

Line 189: Add "K+" here, assuming that the potassium discussed later is from the IC.

Line 235: Define 1:1 v/v notation

Line 239: Replace the IMPROVE convention with a reference, perhaps Pitchford et al., 2007. ("Revised algorithm for estimating light extinction from IMPROVE particle speciation data", JAWMA, 57, 1326-1336).

Line 273: Check notation in table 2 and make sure it is the same as in the text for each species.

Line 295: This is a little confusing. I assume based on Table 2 that the authors are saying $0.1*CM = Al+Fe+Mg$ but it isn't immediately clear from this sentence.

Line 318: Point out that RM is assumed to be organic matter.

Line 342: Define NOx first use

Line 357: Coarse Zn:Al ratios are discussed throughout the paper but from the composition section, it seems like only PM2.5 composition was analyzed. Was the coarse mode speciated also measured?

Line 395: What was the site average? It would be useful to add a column to table 3 with this information for each site.

Line 404: What is the significance of "Study A" and "Study B"? Why are they referred to in this way?

Line 429: There are several instances when the values in the text are not exactly what are reported in the figures. (PM2.5 is 69 in text, 70 in figure)

Line 463: 25% in texts, 24% in figure.

Line 472: 17 in text, 18 in figure.

Line 483: Recommend discussing the sites in the same order as displayed in the figure.

Line 506: PMc notation has not been used previously.

Line 514: 55% in text 59% in figure. 18% in text, 19 in figure, 7% in text, 7.4% in figure.

Line 554: Does total mass here refer to PM2.5?

Line 556: add "respectively" to these comparisons so the reader knows which is which. The order of the comparison switched for CM (note 11% in text, 10% in figure) and EBC.

Line 570: Does "total aerosol mass" here refer to PM2.5?

Line 578: Again PMc notation used here.

Line 592: And the Butler et al value of 55%.

Line 594: CM 10% in text, 11 in figure, ASO4: 21% in text, 24% in figure; ANO3: 3% in text, 3.6% in figure.

Line 610: Replace BC with EBC. Also, 9% in text, 10% in figure.

Line 629: Does this Zn:Al ratio refer to PM2.5 or PM10?

Line 663: What were the average mass scattering efficiencies applied here, and were they consistent with major mass compositions during the same time periods? There are periods with fairly high biases between the mass estimates. Are the assumptions of ∼constant mass and density appropriate during these periods, based on composition data?

Line 678: 0.71 in text, 0.70 in Figure

Tables and Figures

Line 1112: Another reason for switching the order of the hygroscopicity and aerosol

composition sections would be that the species in Table 1 are not defined until Table 2. Switching the order would help to interpret Table 1.

Line 1112: Are the values of PBW averaged across all sites?

Line 1121: There are some discrepancies with notation of species mass in this table and the text. I recommend using "sea salt" instead of "NaCl" since it is used in the text (line 279). Also, 0.18[Na]ss used in the table but 0.18[Na] used in text (line 287). Define RH, X. Define SSR.
* * *

---

## Author Response (AR1)

**Responses to Anonymous Referee #2**

We thank the referee #2 for these helpful comments. Referee comments are numbered, with our responses, and any changes to the manuscript, subsequently given.

1) line 22: "2-22 month periods", what was the average period for all sites? Whether 2 or 22 months of data were available for a given station is highly relevant with regard to its representativeness. Please clarify in the abstract and in the text.

**The text and abstract now note that the average sampling period of sampling is 12 months.**

2) line 34: the term "residue" is rather confusing, especially if one sees later on in the text that it refers mainly to organic matter. I would suggest to find a different term.

**We have changed all mentions of 'residue' to 'residual' matter. We follow C. A. Brock et al.: Characteristics, sources, and transport of aerosols, ACP, 2011, figure 7, which notes of an aerosol "residual = un-speciated mass (probably organic)".**

3) line 51, treatment of outliers: how were outliers processed? This line refers to a major sulphate event, and in the final section of the MS for several stations the authors describe the impact of dust events, major forest fires... How were these outliers dealt with? Depending on the duration of the sampling (see the first comment above), these outliers may have been not representative and strongly impact the mean aerosol concentration. A section should be added to discuss how frequent these outliers were, and how they were treated.

**These major events (e.g. dust storms, annual festivals) affect exposure and are thus included in the averages. These major events are described in detail in Section 5.3. Number of filters and date ranges are included for each site to assess representativeness.**

**For other instances of outlier data, we have added a new section (4.4), which describes major sources of uncertainty for chemical and physical measurements.**

4) lines 116-121: please highlight as a limitation that no data are available for Europe. This is a major spatial gap, despite data being openly available (e.g., chemical speciation data and nephelometry from the EMEP stations, stored in the EBAS database).

**We have added at the lines 169-171:**

**"Site selection prioritizes under-represented globally-dispersed, population-dense regions; no SPARTAN sites yet exist in Europe".**

5) line 135, what was the diameter of the filters, 47 mm?

**Thank you for noting this, we now state the 25mm filter diameter.**

6) line 155: were all the filters from all sites shipped to Dalhousie Univ. for analysis? Please describe how filters were stored + transported to guarantee sample conservation, as this could be a major issue. How many filters/month were sampled, and how many in total per site? Please add a Table (even if in Supplementary material).

**Added 155-161: "As described by Snider et al (2015), loss rates of ammonium nitrate during passive air flow were an order of magnitude less than during active air flow. Thus the sampling protocol is designed to actively sample for one diurnal cycle and to avoid daytime sampling after collecting nighttime PM. Following the IMPROVE protocol (Hand and Malm, 2006), filters are transported at room temperature in sealed containers between measurement sites and the central SPARTAN laboratory at Dalhousie University, where analysis is conducted".**

**The number of filters at each site is stated in Table 3. Line 151-2 states that one filter is sampled every 9 days. Thus there are typically 3 filters/month per site.**

7) line 298: please add a reference for the 0.18 coefficient applied to Na+

**We have added the reference Henning et al. (2003) to this line.**

8) line 320: please rename the fraction "residue matter" and state clearly from the beginning that it refers mainly to organics

**We have modified lines 247-249 to read**

**"Residual matter, which is treated as mainly organics, is estimated by subtracting dry inorganic mass (IN) and its associated water (referenced to our weighing conditions of 35 ± 5 % RH) from total $PM_{2.5}$ mass"**

9) section 4.10: this entire section is very descriptive, but hardly any interpretation of the results is provided.

**We have moved this descriptive overview to section 5.1 to emphasize that it is intended to provide context for interpretation of specific site characteristics in the rest of section 5.**

10) line 357: this result is surprising for Beijing: even despite the influence of dust storms, a major urban area like Beijing should have a high Zn/Al ratio (high anthrop. influence). Whys is this not the case? Could it be due to the potential lack of representativeness of the samples, linked to how outliers were dealt with? Please discuss and clarify in the text.

**We compared our ratios with previous work (Yang et al. 2011), that found a ratio of Zn:Al 0.67, which remains lower than compared with Hanoi and Dhaka. The moderate Zn:Al ratio could reflect the sampling location to the west of the city, upwind of many traffic sources. We have commented further on this in section 5.3.1 (previously 4.12.1), lines 468-473:**

**"The mean $PM_{2.5}$ Zn:Al ratio is lower than other large cities (0.51) likely due to larger fraction of natural dust sources and the sampling location in the northwest quadrant of the city, upwind of many traffic sources. The lowest coarse-mode Zn:Al mass ratios are observed in April 2014 (0.07)**

**and April 2015 (0.06) during the annual Yellow dust storm season. This is balanced by urban dust sources throughout the year, in agreement with Lin et al. (2015) who found evidence of high CM in industrial areas of Beijing".**

11) line 376, 60% in Kanpur: if RM is so high in Kanpur it implies that CM will be very low, which seems surprising for an Indian city. Could this be because sampling took place during the monsoon season, maybe? This refers again to the temporal representativeness of the sampling. Some of the chemical composition results seem unexpected, please include a section discussing the potential sources of uncertainty, e.g. sampling period, filter transport, technical issues during sampling... What could have gone wrong, with such a disperse network of stations?

**The RM in Kanpur is consistent with previous measurements of organics and RM in Kanpur. From section 5.3.6 (lines 566- 569):**

**"Notably the combined OM + unknown fractions from these previous two studies account for two thirds of aerosol mass, 58% for Behera and Sharma (2010) and 63% for Ram et al. (2012), similar to our 59% RM estimate".**

**We also added a new section 4.4 (lines 339-346) to describe our sources of uncertainty regarding loss of semivolatiles:**

**"Of concern is the loss of semivolatiles after sampling. In the laboratory we reduce semivolatile loss by storing filters in closed containers. For time spent in the field, we examined the trend in $PM_{2.5}$ mass and $ANO_3$ from the first filter sampled (54 day residence time in instrument) through the last filter sampled (negligible residence time in instrument). Statistically insignificant trends were found for both $PM_{2.5}$ (-0.09 ± 0.46 µg m$^{-3}$/position) and $ANO_3$ (0.06 ± 0.15 µg m$^{-3}$/position) providing confidence in retention of semivolatiles on filters in the cartridge".**

12) line 394, strong correlation between ASo4 and ANO3: this is unexpected due to the thermal instability of ANO3 in summer (when formation of ASO4 is highest). On an annual basis both components might correlate, but not on a monthly basis. Please clarify.

**We clarified that ammonium and sulfate correlate well, but make no claims that ammonium nitrate correlates well with ammonium sulfate. ($r^2 < 0.15$).**

13) section 4.11: this section is very unclear, what studies are the authors comparing their SPARTAN results with? Please specify. What are "study A" and "Study B" in the Figure?

**We have removed the terms "Study A" and "Study B". The first sentence of this section (lines 433-434) was modified to**

**"We compare SPARTAN $PM_{2.5}$ speciation with one or two previous studies available from the literature (Prior Study in figure 3) and focus on collocated relative $PM_{2.5}$ composition of major species within the last 10 years."**

14) line 417: "expectation that RM is organic", this is not an expectation, it is a definition of the

SPARTAN methodology (given that all other components are already accounted for)

**We modified lines 450-452 to now read:**

**"…SPARTAN measurements of RM appear to be predominantly organic in nature".**

15) line 410: "10-30%", this is an overall limitation of the work: because such a broad range of station types, geographical regions and locations is used, the results also become rather broad and non-specific. E.g., stating that $ASO_4$ contributes with 10-30% to $PM_{2.5}$ mass is not a very specific result, it could represent almost anywhere in the world. Therefore, please state this limitation in the intro or results section, and extract the use of this kind of very global data, e.g. probably (in my opinion) for modelling studies, etc. What is the ultimate purpose of this work? Otherwise it becomes simply a descriptive manuscript.

**The ultimate purpose of work is to describe the initial results about $PM_{2.5}$ concentrations from around the world. Due to the variability of PM components at each location, we provide broad concentration ranges when describing the SPARTAN project as a whole. We are developing modelling studies to further interpret the measured composition.**

**We have modified Lines 18-21 of the Abstract to read:**

**"The Surface PARTiculate mAtter Network (SPARTAN) is a long-term project that includes characterization of chemical and physical attributes of aerosols from filter samples collected worldwide. This manuscript discusses the ongoing efforts of SPARTAN to define and quantify major ions and trace metals found in fine particulate matter ($PM_{2.5}$)".**

**Lines 122-123 in the Introduction now reads:**

**"We discuss the ongoing efforts of the SPARTAN project to quantify major ions and trace metals found in aerosols worldwide".**

16) section 4.12.1: same issue as above, please discuss the relative contribution of natural and anthropogenic dust in Beijing. The large number of samples available (100) should allow for this kind of interpretation.

**We elaborate on the Zn:Al ratio to understand the relative contribution of natural and anthropogenic dust. We are currently preparing a manuscript that characterizes source attribution of different chemicals and elements. Lines 470-475:**

**"The mean PM2.5 Zn:Al ratio is lower than other large cities (0.51) likely due to larger fraction of natural dust sources and the sampling location in the northwest quadrant of the city, upwind of many traffic sources. The lowest coarse-mode Zn:Al mass ratios are observed in April 2014 (0.07) and April 2015 (0.06) during the annual Yellow dust storm season. This is balanced by urban dust sources throughout the year, in agreement with Lin et al. (2015) who found evidence of high CM in industrial areas of Beijing"**

17) line 445: due to volatilzation but also to the fact that the authors are comparing their results with those from different periods in time, in the other studies.

**We now note different sampling periods. Lines 478-480:**

**"SPARTAN ANO$_3$ concentrations (8.5%) are relatively higher than most other locations, though lower than either previous study (11-12 %), possibly due to different sampling periods".**

18) line 520, "organic matter burning": this is another example of a potential outlier. In addition, it would be useful if the authors could add somewhere in the text a brief assessment of major common emission sources, e.g., biomass combustion, agriculture, natural dust... This would help to integrate the results from the different sites rather than simply state ranges of chemical components (e.g., ASO4 = 10-30%) which don't provide much specific information.

**These major common emission sources are described at the start of Section 2. We are preparing a modelling comparison manuscript to more quantitatively assess emission sources.**

19) line 572: festival in Israel, another probable outlier

**We agree this festival is an unusual event. However, it is an event that affects ambient PM. We moved discussion of this event into a separate paragraph.**

20) line 691: black carbon should be equivalent black carbon

**Updated and corrected, thank you.**

21) line 697: "3-year span", please clarify that this is not a continuous monitoring period in all sites, but instead a sequence of consecutive 4-month (approx) periods in different stations. This is a very big difference.

**We have modified this to read**

**"We report ongoing measurements of fine particulate matter (PM$_{2.5}$), including compositional information, in 13 locations in two month or greater intervals all within a three-year span (2013-2016)"**

22) Conclusions: the conclusions section is again very descriptive, it is rather a summary. Actual conclusions and applications for their data should be extracted by the authors.

**We have revised this section to further develop specific conclusions.**

23) As proposed above, a reference to major common emission sources could be added (this is slightly hinted at in paragraph 7369-744).

**We attempt to tailor emission sources to each SPARTAN location. More general land use references include Latham et al (2014) and O:C characteristics from Canagaratna et al. (2015).**

**Responses to Anonymous Referee #3**

We thank the referee #3 for these helpful comments. Referee comments are numbered, with our responses, and any changes to the manuscript, subsequently given.

1)The introduction is a bit disjointed. As a suggestion, I recommend moving the paragraph that begins on line 102 to line 78 (becomes 2nd paragraph), followed by the paragraph starting on line 93, with the sentence starting as "Furthermore, no global. . ." on line 80-82 added to the end, so that the RH effects are included in 1 paragraph. The rest of the paragraph (line 83-91) can become the 4th paragraph, followed by the paragraph at line 114, finishing with the paragraph on line 121. The order of the discussion would then be 1) Health (2) Chemical composition (3) Humidity effects (4) Satellites (5) SPARTAN (6) Purpose

**Thank you, we have adjusted the order of paragraphs to reflect your suggestions.**

2) A difficulty with the current organization of the paper is that the section on hygroscopicity comes before aerosol composition such that aerosol components (like ASO4) are being discussed before the reader knows how the authors define them. It would help to follow the development of the method if the description of the assumed aerosol components came first, and the kappa development followed. I recommend switching the order of section 3 (aerosol hygroscopicity) and section (4) PM2.5 aerosol composition. Also, define a new section for mass speciation results (4.10 and onward). In accord with this reorganizing, switch the order of Figure 1 and 2, and Tables 1 and 2.

**Thank you, we have arranged sections 3 and 4 as you have suggested, and the corresponding tables and figures. We have also moved lines 158-162 to the beginning of the new Section 5.**

*Specific comments follow:*

3) I recommend instead of "ammonium sulfate", referring to ASO4 as "ammoniated sulfate" because the definition doesn't necessarily assume fully neutralized ammonium sulfate (1.375*SO4).

**Thank you, we have made the suggested change throughout text.**

4) Please point out that PM2.5 and PM10 are gravimetrically weighed. Line 19: I'm not sure what "maximize the chemical and physical information" means? It seems like the project is characterizing the chemical and physical attributes of aerosols from filter samples.

**We have removed this sentence to (lines 20-23)**

**"Our methods infer the spatial and temporal variability of $PM_{2.5}$ in a cost-effective manner. Gravimetrically-weighed filters represent multi-day averages of fine particulate matter ($PM_{2.5}$), with a collocated nephelometer sampling air continuously."**

5) Line 24: Define AERONET for first time use.

**Done, this is now referred to Aerosol Robotic Network**

6) Line 28: Consider replacing "baseline" with "background" or "rural/remote".

**We now use your suggestion of using 'background'.**

7) Line 34: What RH?

**We have appended "at 35% RH" to the end of sentence**

8) Line 41: Define IMPROVE for first time use.

**We now refer to IMPROVE as Interagency Monitoring of Protected Visual Environments**

9) Line 42: From the slope, which network had higher mass?

**Our values were higher. Modified sentence to read (lines 42-45)**

**"Comparison of SPARTAN versus coincident measurements from the Interagency Monitoring of Protected Visual Environments (IMPROVE) network at Mammoth Cave yielded…"**

10) Line 51: Change "included" to "including"

**Done**

11) Line 51: What was the standard deviation?

**The standard deviation of kappa is 0.04, which has now been included.**

12) Line 96: Define CSN

**Done. CSN is now defined as Chemical Species Network.**

13) Line 97: Define AIM

**Done. AIM is now defined as the Aerosol Inorganic Model.**

14) Line 99: Define kappa

**We have rephrased lines 102-105 to**

**"More recently Petters and Kreidenweis (2007, 2008, 2013) have developed κ-Kohler theory, which assigns individual hygroscopicity parameters κ to all major components,**

**from insoluble crustal materials to sea-salt.”**

15) Line 126: “As a function of chemical speciation” seems redundant.

**Rephrased lines 126-129 to**

**“Section 3 defines categories of aerosol types (crustal and residue material, black carbon, ammonium nitrate, ammoniated sulfate, sea salt, and trace metal oxides) as a function of specific chemical species.”**

16) Line 134: Provide years of sampling

**Added at line 136-137 “across 13 SPARTAN sites, between 2013 and 2016”**

17) Line 147: Define PM10

**Rephrased lines 149-150 to**

**“Air samples first pass through a bug screen and then a greased impactor plate in order to remove particles larger than 10 μm in diameter”.**

18) Line 157: Is PM2.5 here gravimetric mass or summed constituents?

**Rephrased, and moved to beginning of Section 5 (lines 366):**

**“Gravimetrically-weighed PM$_{2.5}$ concentrations within the period June 2013 to February 2016 span an order of magnitude, from 9 μg m$^{-3}$ (e.g. Atlanta) to nearly 100 μg m$^{-3}$ (Kanpur)”.**

19) Line 166: Add “with other networks”

**Changed lines 169-170 to**

**“The sites of Atlanta and Mammoth Cave are included for instrument inter-comparison purposes with other networks”.**

20) Line 173: This sentence implies that surface reflectance is used to obtain all of the following constituents, not just black carbon.

**Thank you. We corrected lines 177-178 to read**

**“These filters are subsequently analyzed for water-soluble ions, trace metals, and surface reflectance to obtain black carbon”.**

21) Line 189: Add “K+” here, assuming that the potassium discussed later is from the IC.

**Done**.

22) Line 235: Define 1:1 v/v notation

**We have amended lines 295-296 to**

**"A 1:1 volume ratio with water as RH approaches 0% yields a = 2"**

23) Line 239: Replace the IMPROVE convention with a reference, perhaps Pitchford et al., 2007. ("Revised algorithm for estimating light extinction from IMPROVE particle speciation data", JAWMA, 57, 1326-1336).

**Thank you, we have now included this reference on line 299.**

24) Line 273: Check notation in table 2 and make sure it is the same as in the text for each species.

**We now define NaCl as 'Sea Salt' or 'SS' throughout text.**

25) Line 295: This is a little confusing. I assume based on Table 2 that the authors are saying $0.1*CM = Al+Fe+Mg$ but it isn't immediately clear from this sentence.

**Modified lines 222-223 to**

**"we generalize that natural CM is approximately 10x[Al + Fe + Mg]"**

26) Line 318: Point out that RM is assumed to be organic matter.

**Have rephrased lines 247-248 to**

**"Residue matter, which is treated as mainly organics, is estimated by subtracting dry inorganic mass (IN) …"**

27) Line 342: Define NOx first use

**Modified line 383 to "…variation in $NH_3$ and $NO_x$ (NO + $NO_2$) sources"**

28) Line 357: Coarse Zn:Al ratios are discussed throughout the paper but from the composition section, it seems like only PM2.5 composition was analyzed. Was the coarse mode speciated also measured?

**We did measure coarse-mode speciation but focused on fine-mode evaluation for this paper. An upcoming manuscript will contain coarse-mode data.**

29) Line 395: What was the site average? It would be useful to add a column to table 3 with this information for each site.

**We have now provided correlations between ammonium and nitrate in table 3.**

30) Line 404: What is the significance of "Study A" and "Study B"? Why are they referred to in this way?

**Both are now labeled "Prior study".**

31) Line 429: There are several instances when the values in the text are not exactly what are reported in the figures. (PM2.5 is 69 in text, 70 in figure)

**Thank you. We have corrected values to reflect our most recent findings.**

32) Line 463: 25% in texts, 24% in figure.

**Fixed, thank you.**

33) Line 472: 17 in text, 18 in figure.

**Fixed, thank you.**

34) Line 483: Recommend discussing the sites in the same order as displayed in the figure.

**Certain sites did not have comparison figures associated. We aim to order the sites from most to least data available.**

35) Line 506: PMc notation has not been used previously.

**Have included in Methodology section (lines 150-153):**

**Aerosols are collected in sequence on a preweighed Nuclepore filter membrane (8 $\mu$m, SPI) that removes coarse-mode aerosols with diameters from 2.5 - 10 $\mu$m in diameter (PMc), while fine aerosols (PM2.5) are then collected on pre-weighed PTFE filters (2 $\mu$m, SKC).**

36) Line 514: 55% in text 59% in figure. 18% in text, 19 in figure, 7% in text, 7.4% in figure.

**Discrepancies have been fixed, thank you. Concentrations have also been updated reflecting new data acquired since submissions.**

37) Line 554: Does total mass here refer to PM2.5?

**Yes. Modified line 616-617 to read "…and total mass of $PM_{2.5}$ ($r^2$ = 0.76, slope = 1.12)."**

38) Line 556: add "respectively" to these comparisons so the reader knows which is which. The order of the comparison switched for CM (note 11% in text, 10% in figure) and EBC.

**Added the word 'respectively' to lines 617-619**

**"Differences between IMPROVE vs. SPARTAN are small for ASO4 (36% vs. 33%), ANO3 (2.4% vs. 1.2%), CM (7% vs. 11%), and EBC (3.0% vs. 5.6%), respectively".**

39) Line 570: Does "total aerosol mass" here refer to PM2.5?

**Yes. Replaced "total mass" with "total PM2.5 mass".**

40) Line 578: Again PMc notation used here.

**PMc is now defined in methodology section, line 151.**

41) Line 592: And the Butler et al value of 55%.

**Modified lines 632-635 to**

**"SPARTAN component fractions in Atlanta are also consistent with respect to Butler et al. (2003); components CM (12% vs. 10%), ASO4 (23% vs. 28%), ANO3 (3.5% vs 4%) and RM and OM (48% vs 55%) closely match, except for EBC (11% vs. 3%), perhaps reflecting different time periods.".**

42) Line 594: CM 10% in text, 11 in figure, ASO4: 21% in text, 24% in figure; ANO3: 3% in text, 3.6% in figure.

**These discrepancies have been fixed, thank you. Concentrations have also been updated reflecting new data acquired since submissions.**

43) Line 610: Replace BC with EBC. Also, 9% in text, 10% in figure.

**Discrepancy has been fixed, thank you.**

44) Line 629: Does this Zn:Al ratio refer to PM2.5 or PM10?

**Now makes reference to PM2.5.**

45) Line 663: What were the average mass scattering efficiencies applied here, and were they consistent with major mass compositions during the same time periods? There are periods with fairly high biases between the mass estimates. Are the assumptions of constant mass and density appropriate during these periods, based on composition data?

**We tried to clarify at the start of this section that composition-specific $\kappa_v$ values are used.**

46) Line 678: 0.71 in text, 0.70 in Figure

**Fixed, thank you.**

*Tables and Figures*

47) Line 1112: Another reason for switching the order of the hygroscopicity and aerosol composition sections would be that the species in Table 1 are not defined until Table 2. Switching the order would help to interpret Table 1.

**We have changed the order of the tables as suggested.**

48) Line 1112: Are the values of PBW averaged across all sites?

**In table 3, PBW is averaged across sites as a percent of total mass. However, it is reported as absolute mass for individual sites.**

49) Line 1121: There are some discrepancies with notation of species mass in this table and the text. I recommend using "sea salt" instead of "NaCl" since it is used in the text (line 279). Also, 0.18[Na]ss used in the table but 0.18[Na] used in text (line 287). Define RH, X. Define SSR.

**We have now changed NaCl to SS or Sea Salt throughout the text, and added SS subscript. X, SSR, and RH have been defined as well.**

[revised manuscript text omitted]

**Beijing**  $PM_{2.5}$: 69 (3), *n* = 114
8.5 (10)% $ANO_3$,
(12)% $ASO_4$,
2.3 (3.3)% SS,
(14)% CM,
8.8 (5.3)% EBC,
(27)% RM

(Yang et al., 2011) 2005-2006,
OM/OC = 1.7,    $PM_{2.5}$: 119(40)
(7)% $ANO_3$,
(10)% $ASO_4$,
1.3 (0.6)% SS,
(3)% CM,
(5)% EC,
(16)% OM,
(10)% Unk (Oanh et al., 2006) 2001-2004,
OM/OC = 1.7    $PM_{2.5}$: 136 (45)
(1.5)% $ANO_3$,
(1.8)% $ASO_4$,
1.2 (1.2)% SS,
(3)% CM,
(7)% EBC,
(22)% OM,
(24)% Unk

**Bandung**  $PM_{2.5}$: 31 (1), *n* = 77
2.4 (1.4)% $ANO_3$,
(8)% $ASO_4$,
1.0 (0.3)% SS,
8.6 (4.1)% CM,
(4)% EBC,
(19)% RM

(Oanh et al., 2006) 2001-2004.
OM/OC = 2.2,  $PM_{2.5}$: 45.5(10.6),
13(4)% $ANO_3$,
21(3)% $ASO_4$,
1.6(0.2)% SS,
6.6(0.5)% CM,
(4)% EBC,
36(11)% RM

(Lestari and Mauliadi, 2009) 2001- 2007, OM/OC = 2.2   $PM_{2.5}$: 43.5(10.5)
4(6)% $ANO_3$,
4(4)% $ASO_4$,
3(2)% SS,
23(21)% CM,
24(14)% EBC,
42(35)% RM

**Manila**   $PM_{2.5}$: 18 (1), *n* = 63
1.8 (1.2)% $ANO_3$,
(9)% $ASO_4$,
2.9 (2.4)% SS,
(6)% CM,
(19)% EBC,
(21)% RM

(Cohen et al., 2009) 2001-2007,
OM/OC = 2.1,    $PM_{2.5}$: 46 (19),
$ANO_3$ N/A
(9)% $ASO_4$,
0.6 (1.5)% SS,
(1.7)% CM,
(11)% EBC,
57(22)% OM,

**Kanpur** $PM_{2.5}$: 99 (9), *n* = 33
7.4 (5.7)% $ANO_3$,
(13)% $ASO_4$,
0.7 (0.3)% SS,
4.8 (2.9)% CM,
(5.0)% EBC,
(35)% RM

(Behera and Sharma, 2010) Oct. 2007 – Jan 2008,
OM/OC = 2.2,   $PM_{2.5}$: 172 (73),
6.1 (1.3)% $ANO_3$,
(4)% $ASO_4$,
2.6 (0.6)% SS,
(3)% CM,
4.8 (1.1)% EC,
(9)% OM,
(10)% Unk

 (Ram et al., 2012) Dec 2008 – Feb 2009,
OM/OC = 2.2   $PM_{2.5}$: 158 (47)
6.6(4)% $ANO_3$,
(5)% $ASO_4$,
1.5 (0.9)% SS,
(6)% CM*,
(1.1)% EC,
(23)% OM,
(24)% Unk
*Assuming CM = [Ca]/0.034 (Wang, 2015)

**Mammoth Cave NP** $PM_{2.5}$: 13.6 (2), *n* = 19
1.2 (1.0)% $ANO_3$,
(19)% $ASO_4$,
0.8 (0.8)% SS,
(11)% CM,
5.6 (3.2)% EBC,
(34)% RM

(IMPROVE, 2015) June-Aug. 2014,
OM/OC = 2.0,   $PM_{2.5}$: 10.0 (5.8),
2.4 (2.5)% $ANO_3$,
(17)% $ASO_4$,
0.3 (1.6)% SS,
(8)% CM,
(3)% EC,
(30)% OM,
17% Unk+$H_2O$

**Atlanta**    $PM_{2.5}$: 9.1 (1), *n* = 13
3.5 (1.2)% $ANO_3$,
(11)% $ASO_4$,
1.2 (1.2)% SS,
(4.7)% CM,
(2.6)% EBC,
(25)% RM

(Butler et al., 2003) Mar. 1999 –2000 Feb,
OM/OC = 2.0, $PM_{2.5}$: 24.2
(0.2)% $ANO_3$,
(1.0)% $ASO_4$,
(0.8)% CM,
(0.2)% EC,
(5)% OM,

EPA Jan-May (USEPA, 2015), OM/OC = 2.0
$PM_{2.5}$: 8.5
(5)% $ANO_3$,
(15)% $ASO_4$,
1.4 (0.6)% SS
(5)% CM,
9.3 (5)% EC,
(36)% OM,

**Hanoi** $PM_{2.5}$: 39 (4), *n* = 10
4.4 (1.1)% $ANO_3$,
(6)% $ASO_4$,
2.5 (0.6)% SS,
(15)% CM,
(5.8)% EBC,
(22)% RM

(Cohen et al., 2010). 2001 –2008
OM/OC = 2.1,  $PM_{2.5}$: 54 (33)
$ANO_3$ N/A
(20)% $ASO_4$,
0.6 (1.4)% SS
(7)% CM,
(3)% EBC,
(19)% OM,
(2)% Unk + $ANO_3$

[Figure]

**Figure 4: Left Hourly PM$_{2.5}$ estimated from SPARTAN overlaid with a MetOne BAM-1020 (June-August 2014) at the Beijing US Embassy (15 km away). Right: 24-hour SPARTAN PM$_{2.5}$ compared with BAM for the year 2014. Reduced major axis (RMA) slope and Pearson correlations for PM$_{2.5}$ are given in inset.**

**Appendix:**

**Appendix A1**:

**Table A1: Hygroscopicity parameter $\kappa_v$ for various studies on organic material**

| $\kappa_v$ (OM) | Comments | Reference |
|:---:|:---:|:---:|
| **0.045** | Fitted to an aged organic mixture, subsaturated | (Varutbangkul et al., 2006) |
| **0** | IMPROVE network, subsaturated | (Hand and Malm, 2006) |
| **0.10 ± 0.04** | RH > 99%, fitted to SOA precursors | (Prenni et al., 2007) |
| $\mathbf{-0.067 + 0.33}$(O:C) | Fitted, RH > 99% | (Jimenez et al., 2009) |
| $\mathbf{0.29}$(O:C) | RH > 99%, 0.3 < O:C < 0.6 | (Chang et al., 2010) |
| **0.05** | Best estimate from aged mixtures, subsaturated | (Dusek et al., 2011) |
| **0.01 – 0.2** | Field studies & smog chamber, subsaturated | (Duplissy et al., 2011) |
| **0.16** | RH > 99% | (Asa-Awuku et al., 2011) |
| **0.05 – 0.13** | Lab experiments, aged with $H_2O_2$ and light; subsaturated | (Liu et al., 2012) |
| **0.1** | RH > 99%, $D_{dry}$ < 100 nm | (Padró et al., 2012) |
| $\mathbf{0.12\varepsilon_{WSOM}}$[#] | RH > 99% | (Lathem et al., 2013) |
| $\mathbf{-0.005 + 0.19}$(O:C) | Fitted, RH > 99% 100 nm particle | (Rickards et al., 2013) |
| **0.03, 0.1** | HDTMA-measure, subsaturated | (Bezantakos et al., 2013) |
| **0.1** | Subsaturated | Selected for this study |

[#]$\varepsilon_{WSOM}$ = fraction of water-soluble organic material.

**Appendix A2:**

Dry aerosol scatter ($b_{sp,\mathrm{dry}}$) is related to relative humidity (RH) by

$$b_{sp,\mathrm{dry}} = \frac{b_{sp}(\mathrm{RH})}{f_v(\mathrm{RH})} \qquad\qquad \text{Eq. A1}$$

Changes in scatter are also proportional to mass (Chow et al., 2006; Wang et al., 2010)

$$b_{sp,\mathrm{dry}} = \alpha \mathrm{PM}_{2.5,\mathrm{dry}} \qquad\qquad \text{Eq. A2}$$

where $\alpha$ (m$^2$ g$^{-1}$) is the mass scattering efficiency and a function of aerosol size distribution, effective radius, and dry composition. In this study we treat composition, density, and size distribution as constant over each of our 9-day intermittent sampling periods so that $\alpha \approx \langle\alpha\rangle_{9d}$. Under this assumption the predicted mass changes in low humidity (35% RH) are proportional to water-free aerosol scatter:

$$\mathrm{PM}_{2.5,\mathrm{dry}} = \langle\mathrm{PM}_{2.5,\mathrm{dry}}\rangle \frac{b_{sp,\mathrm{dry}}}{\langle b_{sp,\mathrm{dry}}\rangle} \qquad\qquad \text{Eq. A3}$$

where $<>$ indicates 9-day averages. The explicit compensation for aerosol water is then

$$[\mathrm{PM}_{2.5,\mathrm{dry}}] = \frac{\langle[\mathrm{PM}_{2.5,\mathrm{dry}}]\rangle}{\langle b_{sp}(\mathrm{RH})/f_v(\mathrm{RH})\rangle} \cdot \frac{b_{sp}(\mathrm{RH})}{f_v(\mathrm{RH})} \qquad\qquad \text{Eq. A4}$$

where [] indicates concentration in μg m$^{-3}$. Uncertainties are a function of replicate weighing measurements (± 4 μg), flow volume (± 10%), %RH (± 2.5), aerosol scatter (± 5%), and $\kappa_v$ (± 0.05).

$$\left(\frac{\delta[\mathrm{PM}_{2.5,\mathrm{h}}]}{[\mathrm{PM}_{2.5,\mathrm{h}}]}\right)^2 \approx \left(\frac{\delta\mathrm{PM}_{2.5}}{\mathrm{PM}_{2.5}}\right)^2 + \left(\frac{\delta V}{V}\right)^2 + \left(\frac{\delta b_{sp}}{b_{sp}}\right)^2 + \left(\frac{\delta f_v}{f_v}\right)^2 \qquad \text{Eq. A5}$$

where

$$\left(\frac{\delta f_v}{f_v}\right)^2 = \frac{(f_v - 1)^2}{f_v^2}\left[\left(\frac{\delta\kappa}{\kappa}\right)^2 + \left(\frac{\delta\mathrm{RH}}{\mathrm{RH}\cdot(100 - \mathrm{RH})}\right)^2\right] \qquad \text{Eq. A6}$$

The average relative 2-σ PM$_{2.5}$ uncertainty was 26% for dry hourly predictions, increasing with higher RH cutoffs. A cut-off of RH = 80% has been applied to our data, above which hygroscopic uncertainties, as well as total water mass, dominate.

---

## Author Response (AR2)

**Comments for the Author, and replies to editor:**

The concentrations are expressed in μg/m3; however, it is unclear whether the air volumes used were volumes at ambient temperature and pressure or instead at standardized conditions; this should be indicated.

**We have included the sentence at lines 179-180 "Time-integrated flow rates at ambient air pressure and temperature are used to define the sampled volume for aerosol concentrations reported in μg m$^{-3}$".**

Lines 5-14: There are problems with the affiliation of some of the authors. For example, Kebin He is from China and not from Indonesia.
**Thank you. These affiliations have been corrected.**

Line 33: Replace "black carbon" by "equivalent black carbon".
**Done**.

Line 74: Replace "et al. 1994" by "et al., 1994".
**Done**.

Line 84: Replace ", however" by "; however,".
**Done**.

Lines 126-127: Replace "black carbon" by "equivalent black carbon".
**Done**

Line 142: Replace "457nm, 520nm, 634nm" by "457 nm, 520 nm, 634 nm".
**Done**

Line 156: Replace "et al (2015)" by "et al. (2015)".
**Done**

Line 164: Replace "burning," by "burning".
**Done**

Line 179: Replace "black carbon" by "equivalent black carbon".
**Done**

Line 189: First, as appears from Table 1, several more trace metals are measured than Zn, Mg, Fe, and Al; all measured trace metals should be listed here or the text should at least be modified. Secondly, Zn cannot really be called a crustal element. According to Mason (Principles of Geochemistry, 3rd ed., Wiley, 1966) the concentration of Zn in average crustal rock is only 70 ppm; there are more than 20 elements with higher concentrations in average crustal rock.

**Line 191 has been rephrased to "One filter half is analyzed for crustal components Mg, Fe, and Al as well as trace elements Zn, V, Ni, Cu, As, Se, Ag, Cd, Sb, Ba, Ce and Pb."**

Lines 191-193: More information on the ICP-MS analysis is needed; a literature reference should at least be provided.

**This section has been expanded to lines 195-201 "The acid/filter combination is boiled at 97C for 2 hours, and the liquid extract is submitted for quantitative analysis via inductively coupled plasma mass spectrometry (ICP-MS, Thermo Scientific X-Series 2), and follows standardized methodology as in Rice et al. (2012). The ICP-MS analysis is quantified via five concentrations (25, 50, 100, 250, and 500 ug/L) of a 25-element acidified stock solution. Three reference metal ions (45Sc, 115In, and 159Tb) are also used for atomic mass calibration. All ion mass signals are measured in triplicate, and the mean signal value is used for elemental quantification".**

Line 220: Replace "associate as" by "associated as".
**Done**

Line 239: The statement that "Trace elemental oxides are the summation of oxides for all measured ICP trace elements" is incorrect; Mg, Fe and Al are also determined by ICP-MS, but, according to Table 1, not included in TEO.

**Line 247-48 has been rephrased to "Trace elemental oxides are the summation of estimated oxide mass for trace elements as measured by ICP-MS, and make up a negligible portion of total mass (< 1%)."**

Line 325: Reference is made here to Sect. 4.9; however, there is no such section.
**This has been corrected to "Sect. 3.9"**

Line 362: Replace "is ongoing task" by "is an ongoing task".
**Done**

Line 409: Replace "account for" by "accounts for".
**Done**

Line 415: Should it not be "dependent" instead of "independent" here?

**We have attempted to rephrase this sentence to "Although RM, as defined here, is not fully independent from measured ASO$_4$, correlations between these two mass fractions imply related sources".**

Line 417: Replace "potassium K" by "K".
**Done**

Line 456: Replace "relate to" by "relates to".
**Done**

Line 469: Replace "than other" by "than in other".
**Done**

Line 470: Replace "to larger" by "to a larger".
**Done**

Line 480: Replace "than most" by "than for most".
**Done**.

Line 480: Replace "than either" by "than in either".
**Done**

Line 481: Replace "than Yang" by "than in Yang".
**Done**.

Line 482: Replace "than Oanh" by "than in Oanh.
**Done.**

Line 513: Replace "black carbon" by "equivalent black carbon".
**Done**

Line 560: Replace "salt" by "sea salt".
**Done**

Line 568: Replace "for two" by "for almost two".
**Done**

Line 652: Should it not be "Science and Technology" instead of "Science" here?
**Corrected, thank you.**

Line 658: Replace "al.(2010)" by "al. (2010)".
**Done**

Line 667: Replace "10m" by "10 m".
**Done**

Line 687: Replace "orgnic" by "organic".
**Done**

Line 689: Replace "the value" by "with the value".
**Done**

Section 6.2: It is unclear at which RH the PM2.5 data of the Beta Attenuation Monitor were taken. At dry conditions (RH=35%)? If not, how were they converted to dry conditions (RH=35%)? Clarification is definitely needed.
**We have included at line 719 "The BAM instrument contains a drying column with a 35% humidity set point"**

Line 711: Acronyms and abbreviations, here "BAM", should be explained (written full-out) when first used; presumably, "BAM" stands for "Beta Attenuation Monitor".
**This has been now done at line 718.**

Line 718: Replace "panel shows" by "panel in Figure 4 shows".
**Done**

Lines 723-724: The sentence starting with "The agreement" is redundant; it repeats essentially what is already said in line 711.
**We have now deleted this line.**

Line 739: Replace "black carbon" by "equivalent black carbon".
**Done**

Reference list (pages 21-28): Titles of journal articles should be in lower case instead of in Title Case; this comment applies, e.g., to Begum et al. (2012), Behera et al. (2010) and Bell et al. (2007), but also to several more references.
**We have modified the use of capitalization as requested.**

Lines 907-909: This reference should be moved down; it should come after "USEPA, 2014".
**Done**

Line 1001: Replace "Sci. , 326" by "Science, 326".
**Done**

Line 1168, within Table 1: It is stated in column 2 that the measurement method for the CM species is "ICP-MS & IC"; however, none of the three elements listed for CM in column 3 is determined by IC, they are measured by ICP-MS only.
**This had referred to an older method. It is now updated and corrected to ICP-MS only.**

Line 1176: Replace "Residue Matter" by "Residual Matter".
**Done**

Page 30, footnote of Table 3: Replace "Residue Matter" by "Residual Matter".
**Done**

Page 30, footnote of Table 3: It is unclear to what the "Geometric mean of ratio" applies.
**Superscripts "a" and "b" have been added to table. They had been accidentally removed.**

Page 31, caption of Figure 1: The figure shows data for "Ammonium Sulfate" not for "Ammoniated Sulfate" (ASO4). If the figure is correct, the acronym "ASO4" cannot be used in the caption; in addition, the text in lines 289-293 would also need to be revised.

**In the caption of Figure 1, "$ASO_4$" has been replaced with "ammonium sulfate" and "$ANO_3$" has been replaced with "ammonium nitrate".**

**We have also replaced "$ASO_4$" with "ammonium sulfate" at line 301, and included a sentence at line 299-300 "The $\kappa_v$ value for ammonium bisulfate is similar to the $\kappa_v$ value of ammonium sulfate, which is adopted here for $ASO_4$"**

Page 33, line 2 of figure caption: Replace "residual" by "residual matter".
**Done**.

Page 35, within Table A1: Replace "100 nm particle" by "100 nm particles".
**Done**

[revised manuscript text omitted]

8.5 (10)% ANO$_3$,
(12)% ASO$_4$,
2.3 (3.3)% SS,
(14)% CM,
8.8 (5.3)% EBC,
(27)% RM

(Yang et al., 2011) 2005-2006, OM/OC = 1.7,   PM$_{2.5}$: 119(40)

(7)% ANO$_3$,
(10)% ASO$_4$,
1.3 (0.6)% SS,
(3)% CM,
(5)% EC,
(16)% OM,
(10)% Unk (Oanh et al., 2006) 2001-2004, OM/OC = 1.7  PM$_{2.5}$: 136 (45)

(1.5)% ANO$_3$,
(1.8)% ASO$_4$,
1.2 (1.2)% SS,
(3)% CM,
(7)% EBC,
(22)% OM,
(24)% Unk

**Bandung**  PM$_{2.5}$: 31 (1), *n* = 77

2.4 (1.4)% ANO$_3$,
(8)% ASO$_4$,
1.0 (0.3)% SS,
8.6 (4.1)% CM,
(4)% EBC,
(19)% RM

(Oanh et al., 2006) 2001-2004. OM/OC = 2.2,   PM$_{2.5}$: 45.5(10.6),

13(4)% ANO$_3$,
21(3)% ASO$_4$,
1.6(0.2)% SS,
6.6(0.5)% CM,
(4)% EBC,
36(11)% RM

(Lestari and Mauliadi, 2009) 2001- 2007, OM/OC = 2.2   PM$_{2.5}$: 43.5(10.5)

4(6)% ANO$_3$,
4(4)% ASO$_4$,
3(2)% SS,
23(21)% CM,
24(14)% EBC,
42(35)% RM

**Manila**   PM$_{2.5}$: 18 (1), *n* = 63

1.8 (1.2)% ANO$_3$,
(9)% ASO$_4$,
2.9 (2.4)% SS,
(6)% CM,
(19)% EBC,
(21)% RM

(Cohen et al., 2009) 2001-2007, OM/OC = 2.1,   PM$_{2.5}$: 46 (19),

ANO$_3$ N/A
(9)% ASO$_4$,
0.6 (1.5)% SS,
(1.7)% CM,
(11)% EBC,
57(22)% OM,

**Kanpur** PM$_{2.5}$: 99 (9), *n* = 33

7.4 (5.7)% ANO$_3$,
(13)% ASO$_4$,
0.7 (0.3)% SS,
4.8 (2.9)% CM,
(5.0)% EBC,
(35)% RM

(Behera and Sharma, 2010) Oct. 2007 – Jan 2008, OM/OC = 2.2,   PM$_{2.5}$: 172 (73),

6.1 (1.3)% ANO$_3$,
(4)% ASO$_4$,
2.6 (0.6)% SS,
(3)% CM,
4.8 (1.1)% EC,
(9)% OM,
(10)% Unk

 (Ram et al., 2012) Dec 2008 – Feb 2009, OM/OC = 2.2  PM$_{2.5}$: 158 (47)

6.6(4)% ANO$_3$,
(5)% ASO$_4$,
1.5 (0.9)% SS,
(6)% CM*,
(1.1)% EC,
(23)% OM,
(24)% Unk

*Assuming CM = [Ca]/0.034 (Wang, 2015)

**Mammoth Cave NP** PM$_{2.5}$: 13.6 (2), *n* = 19

1.2 (1.0)% ANO$_3$,
(19)% ASO$_4$,
0.8 (0.8)% SS,
(11)% CM,
5.6 (3.2)% EBC,
(34)% RM

(IMPROVE, 2015) June-Aug. 2014, OM/OC = 2.0,   PM$_{2.5}$: 10.0 (5.8),

2.4 (2.5)% ANO$_3$,
(17)% ASO$_4$,
0.3 (1.6)% SS,
(8)% CM,
(3)% EC,
(30)% OM,
17% Unk+H$_2$O

**Atlanta**   PM$_{2.5}$: 9.1 (1), *n* = 13

3.5 (1.2)% ANO$_3$,
(11)% ASO$_4$,
1.2 (1.2)% SS,
(4.7)% CM,
(2.6)% EBC,
(25)% RM

(Butler et al., 2003) Mar. 1999 –2000 Feb, OM/OC = 2.0, PM$_{2.5}$: 24.2

(0.2)% ANO$_3$,
(1.0)% ASO$_4$,
(0.8)% CM,
(0.2)% EC,
(5)% OM,

EPA Jan-May (USEPA, 2015), OM/OC = 2.0 PM$_{2.5}$: 8.5

(5)% ANO$_3$,
(15)% ASO$_4$,
1.4 (0.6)% SS
(5)% CM,
9.3 (5)% EC,
(36)% OM,

**Hanoi** PM$_{2.5}$: 39 (4), *n* = 10

4.4 (1.1)% ANO$_3$,
(6)% ASO$_4$,
2.5 (0.6)% SS,
(15)% CM,
(5.8)% EBC,
(22)% RM

(Cohen et al., 2010). 2001 –2008 OM/OC = 2.1,  PM$_{2.5}$: 54 (33)

ANO$_3$ N/A
(20)% ASO$_4$,
0.6 (1.4)%SS
(7)% CM,
(3)% EBC,
(19)% OM,
(2)% Unk + ANO$_3$

[Figure]

**Figure 4: Left Hourly PM$_{2.5}$ estimated from SPARTAN overlaid with a MetOne BAM-1020 (June-August 2014) at the Beijing US Embassy (15 km away). Right: 24-hour SPARTAN PM$_{2.5}$ compared with BAM for the year 2014. Reduced major axis (RMA) slope and Pearson correlations for PM$_{2.5}$ are given in inset.**

**Appendix:**

**Appendix A1:**

**Table A1: Hygroscopicity parameter $\kappa_v$ for various studies on organic material**

| $\kappa_v$ (OM) | Comments | Reference |
|---|---|---|
| **0.045** | Fitted to an aged organic mixture, subsaturated | (Varutbangkul et al., 2006) |
| **0** | IMPROVE network, subsaturated | (Hand and Malm, 2006) |
| **0.10 ± 0.04** | RH > 99%, fitted to SOA precursors | (Prenni et al., 2007) |
| $-\mathbf{0.067 + 0.33}$(O:C) | Fitted, RH > 99% | (Jimenez et al., 2009) |
| $\mathbf{0.29}$(O:C) | RH > 99%, 0.3 < O:C < 0.6 | (Chang et al., 2010) |
| **0.05** | Best estimate from aged mixtures, subsaturated | (Dusek et al., 2011) |
| **0.01 – 0.2** | Field studies & smog chamber, subsaturated | (Duplissy et al., 2011) |
| **0.16** | RH > 99% | (Asa-Awuku et al., 2011) |
| **0.05 – 0.13** | Lab experiments, aged with $H_2O_2$ and light; subsaturated | (Liu et al., 2012) |
| **0.1** | RH > 99%, $D_{dry}$ < 100 nm | (Padró et al., 2012) |
| $\mathbf{0.12\epsilon_{WSOM}}$[#] | RH > 99% | (Lathem et al., 2013) |
| $-\mathbf{0.005 + 0.19}$(O:C) | Fitted, RH > 99% 100 nm particles | (Rickards et al., 2013) |
| **0.03, 0.1** | HDTMA-measure, subsaturated | (Bezantakos et al., 2013) |
| **0.1** | Subsaturated | Selected for this study |

[#]$\epsilon_{WSOM}$ = fraction of water-soluble organic material.

**Appendix A2:**

Dry aerosol scatter ($b_{sp,\text{dry}}$) is related to relative humidity (RH) by

$$b_{sp,\text{dry}} = \frac{b_{sp}(\text{RH})}{f_v(\text{RH})} \qquad\qquad \text{Eq. A1}$$

Changes in scatter are also proportional to mass (Chow et al., 2006; Wang et al., 2010)

$$b_{sp,\text{dry}} = \alpha \text{PM}_{2.5,\text{dry}} \qquad\qquad \text{Eq. A2}$$

where $\alpha$ ($\text{m}^2\,\text{g}^{-1}$) is the mass scattering efficiency and a function of aerosol size distribution, effective radius, and dry composition. In this study we treat composition, density, and size distribution as constant over each of our 9-day intermittent sampling periods so that $\alpha \approx \langle\alpha\rangle_{9d}$. Under this assumption the predicted mass changes in low humidity (35% RH) are proportional to water-free aerosol scatter:

$$\text{PM}_{2.5,\text{dry}} = \langle \text{PM}_{2.5,\text{dry}} \rangle \frac{b_{sp,\text{dry}}}{\langle b_{sp,\text{dry}} \rangle} \qquad\qquad \text{Eq. A3}$$

where $<>$ indicates 9-day averages. The explicit compensation for aerosol water is then

$$[\text{PM}_{2.5,\text{dry}}] = \frac{\langle [\text{PM}_{2.5,\text{dry}}] \rangle}{\langle b_{sp}(\text{RH})/f_v(\text{RH}) \rangle} \cdot \frac{b_{sp}(\text{RH})}{f_v(\text{RH})} \qquad\qquad \text{Eq. A4}$$

where [] indicates concentration in $\mu\text{g m}^{-3}$. Uncertainties are a function of replicate weighing measurements ($\pm 4\ \mu\text{g}$), flow volume ($\pm 10\%$), %RH ($\pm 2.5$), aerosol scatter ($\pm 5\%$), and $\kappa_v$ ($\pm 0.05$).

$$\left(\frac{\delta[\text{PM}_{2.5,\text{h}}]}{[\text{PM}_{2.5,\text{h}}]}\right)^2 \approx \left(\frac{\delta\text{PM}_{2.5}}{\text{PM}_{2.5}}\right)^2 + \left(\frac{\delta V}{V}\right)^2 + \left(\frac{\delta b_{sp}}{b_{sp}}\right)^2 + \left(\frac{\delta f_v}{f_v}\right)^2 \qquad \text{Eq. A5}$$

where

$$\left(\frac{\delta f_v}{f_v}\right)^2 = \frac{(f_v - 1)^2}{f_v^2}\left[\left(\frac{\delta\kappa}{\kappa}\right)^2 + \left(\frac{\delta\text{RH}}{\text{RH}\cdot(100 - \text{RH})}\right)^2\right] \qquad \text{Eq. A6}$$

The average relative 2-$\sigma$ PM$_{2.5}$ uncertainty was 26% for dry hourly predictions, increasing with higher RH cutoffs. A cut-off of RH = 80% has been applied to our data, above which hygroscopic uncertainties, as well as total water mass, dominate.